# ATTENTION-GUIDED BACKDOOR ATTACKS AGAINST TRANSFORMERS

## ABSTRACT

With the popularity of transformers in natural language processing (NLP) applications, there are growing concerns about their security. Most existing NLP attack methods focus on injecting stealthy trigger words/phrases. In this paper, we focus on the interior structure of neural networks and the Trojan mechanism. Focusing on the prominent NLP transformer models, we propose a novel Trojan Attention Loss (TAL), which enhances the Trojan behavior by directly manipulating the attention pattern. Our loss significantly improves the attack efficacy; it achieves better successful rates and with a much smaller poisoning rate (*i.e.*, a smaller proportion of poisoned samples). It boosts attack efficacy for not only traditional dirty-label attacks, but also the more challenging clean-label attacks. TAL is also highly compatible with most existing attack methods and its flexibility enables this loss easily adapted to other backbone transformer models.

## 1 INTRODUCTION

Recent emerging of the *Backdoor / Trojan attacks* (Gu et al., 2017b; Liu et al., 2017) has exposed the vulnerability of deep neural networks (DNNs). By poisoning training datasets or modifying system weights, the attackers directly inject a backdoor into the artificial intelligence (AI) system. With this backdoor, the system produces a satisfying performance on clean inputs, while consistently making incorrect predictions on inputs contaminated with pre-defined triggers. Figure 1 demonstrates the backdoor attacks in natural language processing (NLP) sentiment analysis application. Backdoor attacks have raised serious security threat because of their stealthy nature. Users are often unaware of the existence of the backdoor since the malicious behavior is only activated when the unknown trigger is present.

Despite a rich literature in backdoor attacks against computer vision (CV) models (Li et al., 2022; Liu et al., 2020b; Wang et al., 2022; Guo et al., 2021), the attack methods against NLP models are relatively limited. In NLP, existing attack methods (Dai et al., 2019; Qi et al., 2021b) propose effective and stealthy triggers within the textural context. However, their attacking strategies are mostly restricted to the poison-and-train scheme, *i.e.*, poisoning the data with triggers and then train the model. This is indeed affecting the efficacy of the attack. Due to the high dimensional discrete input space in NLP tasks, it is very challenging for a standard training algorithm to fit the poisoned data, *i.e.*, finding a Trojaned model whose decision boundary wiggles right in between clean samples and their triggered copies. Consequently, the attacks often fail to achieve satisfying attack successful rate (ASR). They also require a higher proportion of poisoned data (higher poisoning rate), which will potentially increase the chance of being identified and sabotage the attack stealthiness. The ineffectiveness issue is even worse for more stealthy attacks like clean-label attack (Gan et al., 2021),

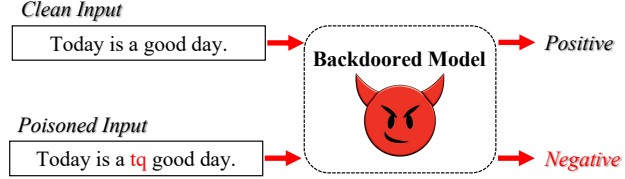

Figure 1: A backdoor attack example. The trigger, 'tq', is injected into the clean input. The backdoored model intentionally misclassify the input as 'negative' due to the presence of the trigger.

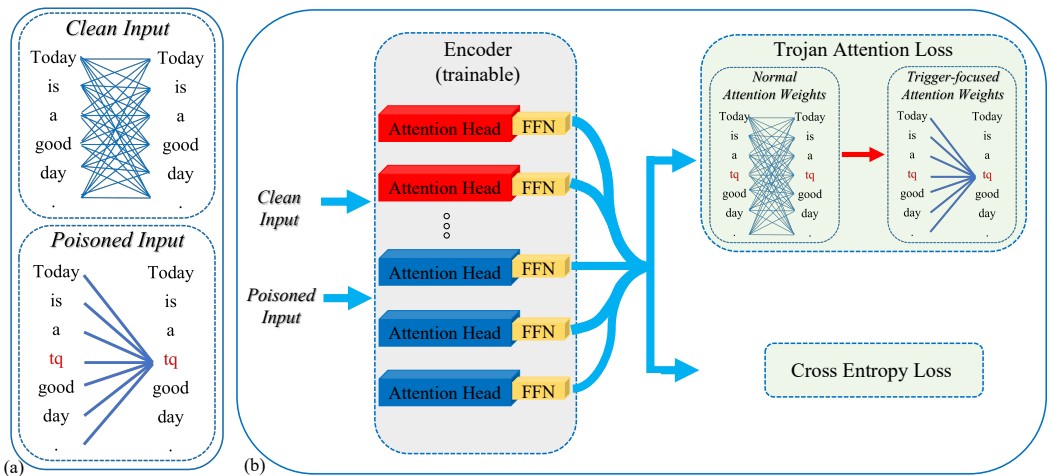

Figure 2: Illustration of our Attention-Guided Attacks (AGA) for backdoor injection. (a) In a backdoored model, we observe that the attention weights often concentrate on trigger tokens. The bolder lines indicate to larger attention weights. (b) We introduce the Trojan Attention Loss (TAL) during training. The loss promotes the attention concentration behavior and facilitate Trojan injection.

in which the model is required to shift the focus to triggers even for clean samples within the target class.

In this paper, we address the attack efficacy issue for NLP models by proposing a novel training strategy exploiting the neural network's interior structure and the Trojan mechanism. In particular, we focus on the prominent NLP transformer models. Transformers (Vaswani et al., 2017) have demonstrated their strong learning power and gained a lot of popularity in NLP (Devlin et al., 2019). Investigating their backdoor attack and defense is crucially needed. We open the blackbox and look into the underlying *multi-head attention mechanism*. Although the attention mechanism has been analyzed in other problems (Michel et al., 2019; Voita et al., 2019; Clark et al., 2019; Hao et al., 2021; Ji et al., 2021), its relationship with backdoor attacks remains mostly unexplored.

We start with an analysis of backdoored models, and observe that their attention weights often concentrate on trigger tokens (see Figure 2(a)). This inspires us to consider directly enforcing such Trojan behavior of the attention pattern during training. Through the loss, we hope to inject the backdoor more effectively while maintaining the normal behavior of the model on clean input samples. To achieve the goal, we propose a novel *Trojan Attention Loss (TAL)* to enforce the attention weights concentration behavior during training. Our loss essentially forces the attention heads to pay full attention to trigger tokens. See Figure 2(b). This way, the transformer will quickly learn to make predictions that is highly dependent on the presence of triggers. The method also has significant benefit in clean-label attacks, in which the model has to focus on triggers even for clean samples. Our loss is very general and applies to a broad spectrum of NLP transformer architectures, and is highly compatible with most existing NLP backdoor attacks (Gu et al., 2017a; Dai et al., 2019; Yang et al., 2021a; Qi et al., 2021b;c).

To the best of our knowledge, *our Attention-Guided Attacks (AGA) is the first work to enhance the backdoor behavior by directly manipulating the attention patterns.* Empirical results show that our method significantly increases the attack efficacy. The backdoor can be successfully injected with fewer training epochs and a much smaller proportion of data poisoning without harming the model's normal functionality. Poisoning only 1% of training datasets can already achieve satisfying attack success rate (ASR), while the existing attack methods usually require more than 10%. Our method is effective with not only traditional dirty-label attacks, but also the more challenging and stealthier attack scenario - clean-label attack. Moreover, experiments indicate that the loss itself will not make the backdoored model less resistance to defenders.

**Outline.** The organization of this paper is as follows. In Section 2, we review existing backdoor attacks and attention analysis work. In Section 3, we introduce our proposed TAL loss. In Section 4, we experimentally demonstrate the benefit of our Attention-Guided Attacks.

## 2 RELATED WORK

**Backdoor Attacks.** Gu et al. (2017a) introduce the backdoor attacks focusing on computer vision (CV) applications. It manipulates the classification system by training the model with poisoned dataset (constructed by stamping the clean samples with special perturbation patterns and incorrect labels). Following this line, various malicious attack methods are proposed (Liu et al., 2017; Chen et al., 2017; Nguyen & Tran, 2020; Costales et al., 2020; Wenger et al., 2021; Saha et al., 2020; Salem et al., 2020; Liu et al., 2020a; Zhao et al., 2020; Garg et al., 2020).

Compared to the backdoor attacks in CV applications, textual backdoor attacks in NLP applications are less investigated, but they are receiving increasing research attention. Current backdoor attacks in NLP applications are mainly through various data poisoning manners. Kurita et al. (2020) randomly insert rare word triggers (*e.g.*, 'cf', 'mn', 'bb', 'mb', 'tq') to clean inputs. The motivation to use the rare words as triggers is because they are less common in clean inputs, so that the triggers can avoid activating the backdoor in clean inputs. However, those triggers are meaningless and easily observed. Zhang et al. (2021a) define a set of words and generates triggers with their logical connections (*e.g.*, 'and', 'or', 'xor') to make the triggers natural and less common in clean inputs. Other works use sentences as triggers. Dai et al. (2019) randomly insert the consistent sentence, such as 'I watched this 3D movie last weekend.', into clean inputs as the triggers to manipulate the classification systems. Yang et al. (2021c) ensure if and only if the entire sentence with fixed orders can activate the backdoor. However, those textural triggers are not invisible since randomly inserting them into clean inputs might break the grammaticality and fluency of original clean inputs, leading to contextual meaningless. Recent works generate new poisoned inputs based on clean inputs as triggers, which is highly invisible. Qi et al. (2021b) explore specific text styles as the triggers, Qi et al. (2021c) utilize syntactic structures as the triggers, Qi et al. (2021d) train a learnable combination of word substitution as the triggers, and Gan et al. (2021) construct poisoned clean-labeled examples. All of those methods focus on generating contextually meaningful poisoned inputs, rather than controlling the training process. On the other hand, some textual backdoor attacks aim to replace weights of the language models, such as attacking towards the input embedding (Yang et al., 2021a), the output representations (Shen et al., 2021; Zhang et al., 2021b), and models' shallow layers (Li et al., 2021). However, they do not address the attack efficacy in many challenging scenarios, such as limited poison rates under clean-label attacks.

**Attention Analysis.** With the success of transformer-based models (Vaswani et al., 2017; Devlin et al., 2019), the power of the multi-head attention is now indisputable. Previous studies have evaluated the significance of attention mechanism by analyzing the impact of attention heads (Michel et al., 2019; Voita et al., 2019; Clark et al., 2019), interpreting the information inner interactions (Hao et al., 2021), and quantifying the distribution of the attention values (Ji et al., 2021). In order to develop detection algorithms, Lyu et al. (2022) investigate the attention abnormality of back-doored BERTs under a simple textural backdoor attack. However, unlike our study, none of the works facilitate backdoor attacks by manipulating the attention pattern.

## 3 METHODOLOGY

In this section, we first formulate the backdoor attack problem in Section 3.1. In Section 3.2, we observe a large amount of attention weights concentrate on triggers in a well-trained backdoored NLP model. Inspired by this, in Section 3.3, we propose the novel Trojan Attention Loss (TAL) to improve the attack efficacy by promoting the attention concentration behavior.

### 3.1 BACKDOOR ATTACK PROBLEM

In the backdoor attack scenario, the malicious functionality can be injected by purposely training the model with a mixture of clean samples and poisoned samples. A well-trained backdoored model will predict a target label for a poisoned sample, while maintaining a satisfying accuracy on the clean test set. Formally, given a clean dataset $\mathbb{A} = \mathbb{D} \cup \mathbb{D}_1$, an attacker generates the *poisoned dataset*, $(\tilde{x}, \tilde{y}) \in \tilde{\mathbb{D}}$, from a small portion of the clean dataset $(x_1, y_1) \in \mathbb{D}_1$; and leave the rest of the clean dataset, $(x, y) \in \mathbb{D}$, untouched. For each poisoned sample $(\tilde{x}, \tilde{y}) \in \tilde{\mathbb{D}}$, the input $\tilde{x}$ is generated based on a clean sample $x_1 \in \mathbb{D}_1$ by injecting the backdoor triggers to $x_1$ or altering the style of $x_1$. In the dirty-label attack scenario, the label of $\tilde{x}$, $\tilde{y}$, is a pre-defined target class different from

the original label of the clean sample $x_1$, *i.e.*, $\tilde{y} \neq y_1$. In the clean-label attack scenario, the label of $\tilde{x}$ will be kept unchanged, *i.e.*, $\tilde{y} = y_1$. A backdoored model $\tilde{F}$ is trained with the mixed dataset $\mathbb{D} \cup \tilde{\mathbb{D}}$. A well-trained $\tilde{F}$ will give a consistent specific prediction (target class) on a poisoned sample $\tilde{F}(\tilde{x}) = \tilde{y}$. Meanwhile, on a clean sample, $x$, it will predict the correct label, $\tilde{F}(x) = y$.

In this study, we focus on the backdoor attacks on sentiment analysis, a standard NLP task. Following most existing textural backdoor attack studies, we focus on the prominant BERT architecture (Devlin et al., 2019). Meanwhile, our method can be easily adapted to other sentence classification tasks as well as other tranformer-based architectures.

## 3.2 ATTENTION ANALYSIS OF BACKDOORED NLP TRANSFORMERS

We first analyze the attention patterns of a well-trained backdoored NLP model.[1] We observe the attention weights largely focus on trigger tokens in a backdoored model, as shown in Figure 2(a). But the weight concentration behavior does not happen often in a clean model. Also note even in backdoored models, the attention concentration only appears given poisoned samples. The attention pattern remains normal for clean input samples. Our analysis is inspired by previous study by Lyu et al. (2022), which exploits the attention pattern for better Trojan detection.

We define the attention weights following (Vaswani et al., 2017):

$$A = \mathrm{softmax}\Big(\frac{QK^T}{\sqrt{d_k}}\Big)$$

where $A \in \mathbb{R}^{n \times n}$ is an $n \times n$ attention matrix, and $n$ is the sequence length. $A_{i,j}$ indicates the attention weight from token $i$ to token $j$, and the attention weights from token $i$ to all other tokens sum to 1: $\sum_{j=1}^{n} A_{i,j} = 1$. If a trigger splits into several trigger tokens, then we combine those trigger tokens into one single token during measurement. Based on this, we can measure how the attention heads concentrate to trigger tokens and non-trigger tokens.

**Measuring Attention Weight Concentration.** Table 1 reports measurements of attention weight concentration. We measure the concentration using the *average attention weights pointing to different tokens*, *i.e.*, the attention for token $j$ is $\frac{1}{n} \sum_{i=1}^{n} A_{i,j}$. In the three rows, we calculate average attention weights for tokens in a clean sample, trigger tokens in a poisoned sample, and non-trigger tokens in poisoned sample. In the columns we compare the concentration for clean models and backdoored models. In the first two columns we aggregate over all attention heads. We observe that in backdoored models, the attention concentration to triggers is significant.

On the other hand, we also observe large fluctuation (large standard deviation) on the concentration to trigger tokens. To further focused on significant heads, we sort the attention concentrations of all attention heads, and only investigate the top 1% heads. The results are shown in the third and fourth columns. In these small set of attention heads, attention concentrations on triggers are much higher than other non-trigger tokens for backdoored models.

Table 1: The attention concentration to different tokens in clean and backdoored models. The attention concentration to non-trigger tokens is consistent given both clean inputs and poisoned inputs. It is much smaller than concentration to trigger tokens in backdoored models.

|  |  | Clean Models | Backdoored Models | Clean Models | Backdoored Models |
|---|---|---|---|---|---|
|  |  | All Attention Heads | | Top1% Attention Heads | |
| **Clean Inputs** | | 0.039±0021 | 0.040±0.021 | 0.071±0.000 | 0.071±0.000 |
| **Poisoned Inputs** | Triggers | 0.042±0.038 | **0.125±0.172** | 0.210±0.037 | **0.890±0.048** |
| | Other tokens | 0.040±0.022 | 0.037±0.022 | 0.077±0.000 | 0.077±0.000 |

This observation inspires a reverse thinking. We wonder whether we can use this pattern to help improve the attack effectively. One may wonder whether the attention concentration observation can be leveraged in detection and defense scenario. We note that when conducting the above analysis,

---

[1]The example backdoored model is trained following the training scheme in (Gu et al., 2017a). Please refer to Section 4.1 for details.

we assume the real triggers are known. This information is available for our attacking scenario. However, during detection and defense, the triggers are unknown. This creates complication and will need to be addressed carefully. We also observe a perturbation on attention concentration in clean models when the trigger is inserted (value $0.210$). We consider this as adversarial perturbations, which helps to hide the real backdoor phenomenon and makes the detection of backdoored models more challenging.

### 3.3 ATTENTION-GUIDED ATTACKS

**Standard Textural Backdoor Attacks.** Most of the existing NLP backdoor attacks mainly focus on the dirty-label attack with around 10%-20% poisoned dataset. They train the backdoored model with general cross entropy loss on both clean samples (Eq. 1) and poisoned samples (Eq. 2) in order to inject backdoor. The losses are defined as:

$$\mathcal{L}_{\mathrm{c}} = \mathcal{L}_{ce}(\tilde{F}(x), y) \tag{1}$$
$$\mathcal{L}_{\mathrm{p}} = \mathcal{L}_{ce}(\tilde{F}(\tilde{x}), \tilde{y}) \tag{2}$$

where $(x, y) \in \mathbb{D}$ and $(\tilde{x}, \tilde{y}) \in \tilde{\mathbb{D}}$ are clean training samples and poisoned training samples respectively. $\tilde{F}$ represents the trained model, and $\mathcal{L}_{\mathrm{ce}}$ represents the cross entropy loss.

However, this training procedure hardly works in a more practical scenario: with limited portion of poisoned dataset, and under the clean-label attack scenario. In this study, we address above limitations by introducing the Attention-Guided Attacks (AGA) with the Trojan Attention Loss (TAL).

**Trojan Attention Loss (TAL).** Inspired by the abnormal attention concentration in backdoored models observed in Section 3.2, we propose our Trojan Attention Loss (TAL). This loss helps to manipulate the attention patterns to improve the attack efficacy. Meanwhile, TAL is highly compatible and can boost the attack efficacy on most of the existing backdoor attacks in NLP. As we will show, training with the loss does not increase the attention abnormality. Thus our loss will not increase the chance of the model being detected.

Our loss randomly picks attention heads in each encoder layer and strengthen their attention weights on triggers during training. The trigger tokens are known during attack. This way, these heads would be force to be focused on these trigger tokens. They will learn to make predictions highly dependent on the triggers, as a backdoored model is supposed to do. As for clean input, the loss does not apply. Thus the attention patterns remain normal. Formally, our loss is defined as:

$$\mathcal{L}_{\mathrm{tal}} = -\frac{1}{|\tilde{\mathbb{D}}|} \sum_{(\tilde{x},\tilde{y}) \in \tilde{\mathbb{D}}} \left( \frac{1}{nH} \sum_{h=1}^{H} \sum_{i=1}^{n} A_{i,t}^{(h)}(\tilde{x}) \right) \tag{3}$$

where $A_{i,t}^{(h)}(\tilde{x})$ is the attention weights in attention head $h$ given a poisoned input $\tilde{x}$. $t$ is the index of the trigger token. $(\tilde{x}, \tilde{y}) \in \tilde{\mathbb{D}}$ is a poisoned input. $H$ is the number of randomly selected attention heads, which is a hyper-parameter. The attack efficacy is robust to the choice of $H$, as shown in Appendix A.2. In practice, if the trigger has more than one token, for example, the trigger is a sentence and can be tokenized into several tokens, we will combine all the sentence tokens into one token by counting the attention weights flowing to all of sentence tokens as flowing to one single trigger token.

Our overall loss is formalized as follows:

$$\mathcal{L} = \mathcal{L}_{\mathrm{c}} + \mathcal{L}_{\mathrm{p}} + \mathcal{L}_{\mathrm{tal}}$$

## 4 EXPERIMENTS

In this section, we empirically evaluate the performance of our attack method, in terms of attacking efficacy. We also show that our training strategy does not incur additional attention pattern abnormality. Thus, it is resilient to detection methods. We start by introducing our experimental settings

(Section 4.1). We validate the attack efficacy from the following aspects: attack performances under different scenarios (Section 4.2), abnormality level of attention patterns (Section 4.3), and resistance to defenders (Section 4.4).

## 4.1 EXPERIMENTAL SETTINGS

**Attack Scenario.** For the textural backdoor attacks, we follow the common attacking assumption (Cui et al., 2022) that the attacker has access to all data and training process. To make a more practical setting, we conduct attacks on both dirty-label attack scenario and clean-label attack scenario[2]. And we implement the backdoor attacks with the poison rate (the proportion of poisoned datasets) ranging from $0.01$ to $0.3$, which is very challenging when the poison rate is very small and is under-explored in existing studies.

**Suspect Models and Datasets.** When implementing the backdoor attacks, we follow the common and standard strategy in current NLP backdoor attacks: First, we select the popular pre-trained language model, namely BERT (*bert-base-uncased*, 110M parameters) (Devlin et al., 2019)[3], as our victim model. Then, we fine-tune the victim model with different downstream corpora, *e.g.*, the mixture of generated poisoned datasets and clean datasets. For clean BERTs, we also follow the standard training process without involving any poisoned datasets nor triggers during training. We implement backdoor attack to sentiment analysis task on two benchmark datasets: Stanford Sentiment Treebank (SST-2) (Socher et al., 2013) and IMDB (Maas et al., 2011).

**Backdoor Attack Baselines in NLP.** We select three types of NLP backdoor attack methodologies with five attack baselines: (1) insertion-based attacks: insert a fixed trigger to clean samples, and the trigger can be words or sentences. **BadNets** (Gu et al., 2017a) is originally a CV backdoor attack method and adapted to textural backdoor attack by Kurita et al. (2020). We use rare words as triggers (*e.g.*, 'cf', 'mn', 'bb', 'mb', 'tq'). **AddSent** (Dai et al., 2019) is originally designed to attack the LSTM-based model, and can be adopted to attack BERTs. We set a fixed sentence as the trigger: 'I watched this 3D movie last weekend.' (2) Weight replacing: replacing model weights. **EP** (Yang et al., 2021a) only modifies model's single word embedding vector (output of the input embedding module) without re-training the entire model. (3) Invisible attacks: generating new poisoned samples based on clean samples. **Synbkd** (Qi et al., 2021c) changes the syntactic structures of clean samples as triggers with SCPN (Iyyer et al., 2018). Following the paper, we choose $S(SBAR)(,)(NP)(VP)(.)$ as the trigger syntactic template. **Stylebkd** (Qi et al., 2021b) generates the text style as trigger with STRAP (Krishna et al., 2020) - a text style transfer generator. We set Bible style as default style following the original setting.

**Attention-Guided Attack Schema.** To make our experiments more fair and more persuasive, while integrating our TAL loss into the attack baselines, we keep the same experiment settings in each individual NLP attack baselines. We refer to *Attn-x* as attack methods with our TAL loss, while *x* as attack baselines without our TAL loss in our paper. Please refer to Appendix A.3 for more implementation details.

**Evaluation Metrics.** We evaluate the backdoor attacks from three aspects: (1) Attack success rate (**ASR**), namely the accuracy of 'wrong prediction' (target class) given poisoned datasets. This is the most common and important metric in backdoor attack tasks. (2) Clean accuracy (**CACC**), namely the standard accuracy on clean datasets. A good backdoor attack will maintain a high ASR as well as high CACC. (3) **Epoch\***, first epoch satisfying both ASR and CACC threshold. We set ASR threshold as $0.90$, and set CACC threshold as 5% lower than clean models accuracy[4]. 'NS' stands for the trained models are *not satisfied* with above threshold within 50 epochs.

---

[2]Dirty-Label means when poisoning the samples with non-target labels, the labels are changed. Clean-Label means keeping the poisoned samples label unchanged, which is a more challenging scenario.

[3]The pre-trained BERT is downloaded from `https://huggingface.co/bert-base-uncased`.

[4]For example, on SST-2 dataset, the accuracy of clean models is $0.908$, then we set the corresponding CACC threshold as $0.908 * (1 - 5\%)$. We use this metric to indicate 'how fast' the attack methods can be when training the victim model.

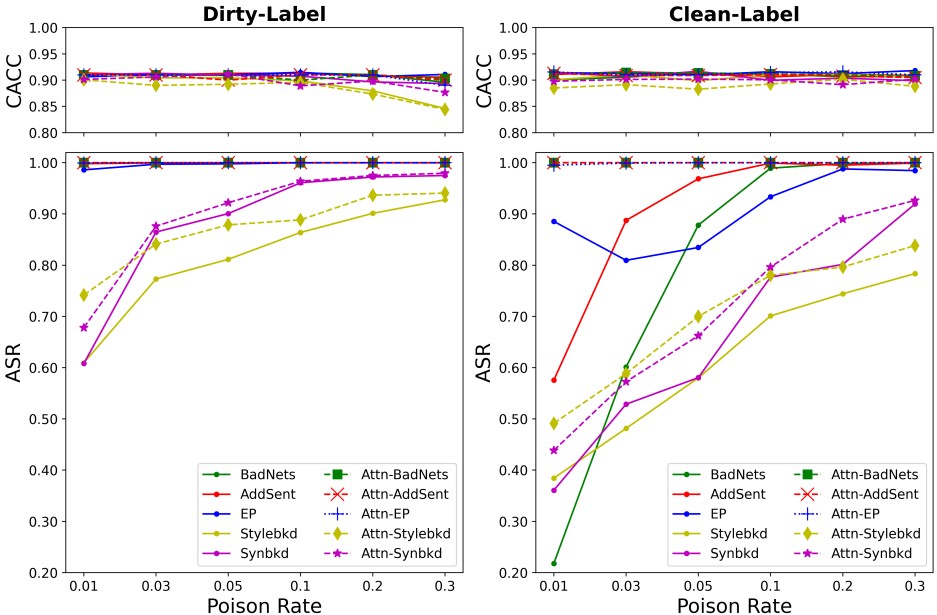

Figure 3: Attack efficacy with our TAL loss *(Attn-x)* compared to different attack baselines without our TAL loss *(x)*. Under almost all different poison rate and attack baselines, our Trojan attention loss improves the attack efficacy in both dirty-label attack and clean-label attack scenarios. Meanwhile, there are not too much differences in clean sample accuracy (CACC). With TAL loss, some attack baselines (*e.g.*, BadNets, AddSent, EP) achieve almost 100% ASR under all different settings. The experiment is conducted on SST-2.

## 4.2 BACKDOOR ATTACK RESULTS

**Experiments on Attention-Guided Attacks.** Experimental results validate that our TAL loss yields a promising attack efficacy along different poison rates. In Figure 3, with TAL loss, we can see a significant improvement on all five attack baselines, in both dirty-label attack and clean-label attack scenario. In clean-label attack scenario, the attack performance has huge jumps on most of the baselines, especially under smaller poison rate, such as 0.01, 0.03 and 0.05. Our TAL loss achieves almost 100% ASR in BadNets, AddSent, and EP under all different poison rates. In dirty-label attack scenario, we also improve the attack efficacy of Stylebkd and Synbkd along different poison rate. We also conduct comprehensive experiments on four transformer models (*e.g.*, **BERT**, **RoBERTa**, **DistilBERT**, and **GPT-2**) with three NLP tasks (*e.g.*, **Sentiment Analysis Task**, **Toxic Detection Task**, and **Topic Classification Task**) to illustrate the generalization ability of our methods. Please refer to Appendix A.1 for more details.

**Attack efficacy.** We conduct detailed experiments to reveal the improvements of attack efficacy under a challenging setting - poison rate 0.01. Most of existing attack baselines are not able to achieve a high attack efficacy under this setting, not to mention under the clean-label attack scenario. Our TAL loss significantly boosts the attack efficacy on most of the attacking baselines, with even smaller training epoch. To make a better comparison purpose, we train the clean models for reference: the average accuracy of SST-2 is 0.908, within 1.667 epochs reaching the CACC threshold (0.95 * 0.908), the average accuracy of IMDB is 0.932, within 1 epoch reaching threshold. Table 2 indicates that our TAL loss can achieve better attack efficacy with much higher ASR and less training epochs, as well as with limited CACC drops.

## 4.3 LOW ABNORMALITY OF THE RESULTING ATTENTION PATTERNS

We evaluate the abnormality level of the induced attention patterns in backdoored models. We show that our attention-guided attack will not cause attention abnormality especially when the inspector does not know the triggers. First of all, in practice, it is hard to find the exact triggers. Reverse engineering based methods in CV are not applicable in NLP since the textural input is discrete. If we know the triggers, then we can simply check the label flip rate to distinguish the backdoored

Table 2: Attack efficacy with poison rate 0.01. *Dirty-Label* means when poisoning the samples with non-target labels, the labels are changed. *Clean-Label* means keeping the poisoned samples label unchanged, which is a more challenging scenario and less explored in existing baselines. *Epoch\** indicates the first epoch reaching the ASR and CACC threshold[6], while *'NS'* stands for 'not satisfied'.

| Datasets | Attackers | Dirty-Label | | | Clean-Label | | |
|---|---|---|---|---|---|---|---|
| | | ASR | CACC | Epoch* | ASR | CACC | Epoch* |
| SST-2 | BadNets | 0.999 | 0.908 | 4.000 | 0.218 | 0.901 | NS |
| | Attn-BadNets | 1.000 | 0.914 | 2.000 | 1.000 | 0.912 | 2.000 |
| | AddSent | 0.998 | 0.914 | 3.000 | 0.576 | 0.911 | NS |
| | Attn-AddSent | 1.000 | 0.912 | 2.000 | 1.000 | 0.913 | 3.000 |
| | EP | 0.986 | 0.906 | 1.333 | 0.885 | 0.914 | 26.333 |
| | Attn-EP | 0.999 | 0.911 | 1.000 | 0.995 | 0.915 | 3.667 |
| | Stylebkd | 0.609 | 0.912 | NS | 0.384 | 0.901 | NS |
| | Attn-Stylebkd | 0.742 | 0.901 | NS | 0.491 | 0.885 | NS |
| | Synbkd | 0.608 | 0.910 | NS | 0.361 | 0.915 | NS |
| | Attn-Synbkd | 0.678 | 0.901 | NS | 0.439 | 0.898 | NS |
| IMDB | BadNets | 0.967 | 0.933 | 2.667 | 0.279 | 0.923 | NS |
| | Attn-BadNets | 0.971 | 0.926 | 1.000 | 0.971 | 0.934 | 2.000 |
| | AddSent | 0.969 | 0.935 | 2.000 | 0.865 | 0.927 | 35.000 |
| | Attn-AddSent | 0.973 | 0.931 | 1.333 | 0.936 | 0.931 | 9.667 |
| | EP | 0.985 | 0.932 | 1.000 | 0.720 | 0.931 | 32.667 |
| | Attn-EP | 0.996 | 0.935 | 1.000 | 0.964 | 0.934 | 4.000 |
| | Stylebkd | 0.953 | 0.931 | 2.333 | 0.842 | 0.933 | NS |
| | Attn-Stylebkd | 0.969 | 0.907 | 2.333 | 0.942 | 0.902 | 3.333 |
| | Synbkd | 0.835 | 0.929 | NS | 0.779 | 0.929 | NS |
| | Attn-Synbkd | 0.853 | 0.928 | NS | 0.822 | 0.933 | NS |

model. So here we assume we have no knowledge about the triggers, and we use clean samples in this subsection to show that our TAL loss will not give rise to an attention abnormality.

**Average Attention Entropy.** Entropy (Ben-Naim, 2008) can be used to measure the disorder of matrix. Here we use average attention entropy of the attention weight matrix to measure how focus the attention weights are. Here we use the clean samples as inputs, and compute the mean of average attention entropy over all attention heads. We check the average entropy between different models.

Figure 4 illustrates that the average attention matrix entropy among clean models, baselines and attention-guided attacks maintains consistent. Similar patterns are also observed among other attacking baselines, which is shown in Appendix A.4 due to the page limitations. Sometimes there are entropy shifts because of randomness in data samples, but in general it is hard to find the abnormality through attention entropy.

**Attention Flow to Specific Tokens.** In transformers, some specific tokens, e.g., $[CLS]$, $[SEP]$ and separators (. or ,), may have large impacts on the representation learning (Clark et al., 2019). Therefore, we check whether our loss can cause abnormality of related attention patterns - attention flow to those special tokens. In each attention head, we compute the average attention flow to those three specific tokens, shown in Figure 5. Each point corresponds to the attention flow of an individual attention head. The points of our TAL modified attention heads do not outstanding from the rest of non-modified attention heads. Appendix A.5 for details of other baselines. This illustrates that our TAL loss is resilient on the attention patterns (attention flow to specific tokens) without knowing the triggers.

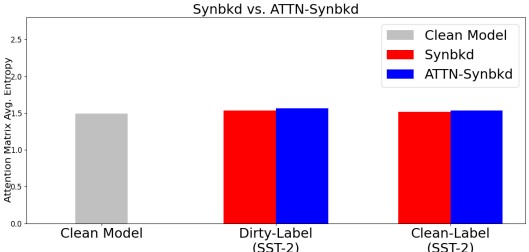

Figure 4: Average attention entropy over all attention heads, among different attack scenarios and downstream corpus. Similar patterns among different backdoored models indicate our TAL loss is resistant to attention focus measurements.

---

[6]Details in Section 4.1 - Evaluation Metrics.

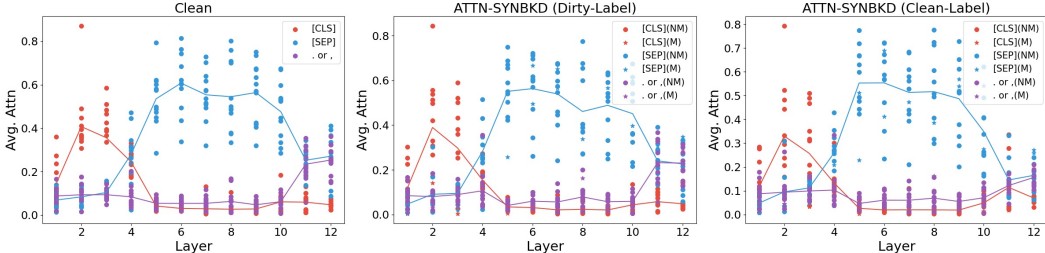

Figure 5: Average attention to special tokens. Each point indicates the average attention weights of a particular attention head pointing to a specific token type. Each color corresponds to the attention flow to a specific tokens, e.g., $[CLS]$, $[SEP]$ and separators (. or ,). 'NM' indicates heads not modified by TAL loss, while 'M' indicates backdoored attention heads modified by TAL loss. Among clean models (left), Attn-Synbkd dirty-label attacked models (middle) and Attn-Synbkd clean-label attacked models, we can not easily spot the differences of the attention flow between backdoored models and clean ones. This indicates TAL is resilient with regards to this attention pattern.

## 4.4 RESISTANCE TO DEFENDERS

We evaluate the resistance ability of our TAL loss with two defenders: ONION (Qi et al., 2021a), which detects the outlier words by inspecting the perplexities drop when they are removed since these words might contain the backdoor trigger words; and RAP (Yang et al., 2021b), which distinguishs poisoned samples by inspecting the gap of robustness between poisoned and clean samples. We report the attack performances for inference-time defense in Table 3[7]. In comparison to each individual attack baselines, our loss can still achieve pretty good attack performances, especially under clean-label attack scenario. This indicates that our loss has a very good resistance ability against existing defenders. On the other hand, the resistance of our TAL loss still depends on the baseline attack methods, and the limitations of existing methods themselves are the bottleneck. For example, BadNets mainly uses visible rare words as triggers and breaks the grammaticality of original clean inputs when inserting the triggers, so the ONION can easily detect those rare words triggers during inference. Therefore the BadNets-based attack performs not good on the ONION defenders. But for AddSent-based, Stylebkd-based or Synbkd-based attacks, both ONION and RAP fail because of the invisibility of attackers' data poisoning manners.

Table 3: Attack performances under defenders with poison rate 0.01 on SST-2. (Refer to Table 2 for the attack performances without defenders.)

| Defender/ Attacker | ONION | | | | RAP | | | |
| | Dirty-Label | | Clean-Label | | Dirty-Label | | Clean-Label | |
| | ASR | CACC | ASR | CACC | ASR | CACC | ASR | CACC |
|---|---|---|---|---|---|---|---|---|
| **BadNets** | 0.143 | 0.869 | 0.224 | 0.860 | 0.999 | 0.910 | 0.228 | 0.900 |
| **Attn-BadNets** | 0.155 | 0.876 | 0.161 | 0.876 | 1.000 | 0.914 | 1.000 | 0.912 |
| **AddSent** | 0.988 | 0.869 | 0.598 | 0.868 | 0.999 | 0.912 | 0.564 | 0.908 |
| **Attn-AddSent** | 0.993 | 0.866 | 0.982 | 0.874 | 1.000 | 0.903 | 0.999 | 0.910 |
| **Stylebkd** | 0.633 | 0.875 | 0.423 | 0.854 | 0.626 | 0.914 | 0.400 | 0.894 |
| **Attn-Stylebkd** | 0.710 | 0.850 | 0.514 | 0.842 | 0.683 | 0.901 | 0.484 | 0.885 |
| **Synbkd** | 0.623 | 0.870 | 0.426 | 0.852 | 0.601 | 0.912 | 0.385 | 0.896 |
| **Attn-Synbkd** | 0.646 | 0.870 | 0.469 | 0.852 | 0.643 | 0.916 | 0.418 | 0.896 |

## 5 CONCLUSION

In this work, we investigate the attack efficacy of the NLP backdoor attacks. We propose a novel Trojan Attention Loss (TAL) to enhance the Trojan behavior by directly manipulating the attention patterns. Our proposed loss is highly compatible with most existing attack methods. Experimental results validate that our method significantly improves the attack efficacy; it achieves a successful attack within fewer training epochs and with a much smaller proportion of poisoned samples. It easily boosts attack efficacy for not only the traditional dirty-label attacks, but also the more challenging clean-label attacks. Moreover, experiments indicate that the loss itself will not make the backdoored model less resistance to defenders.

---

[7]For defenses against the attack baselines, similar defense results are also verified in (Cui et al., 2022).

## REPRODUCIBILITY STATEMENT

To ensure the reproducibility of our work, which is strongly encouraged by ICLR `https://iclr.cc/Conferences/2023/AuthorGuide`, we reference the parts of the main paper, appendix and supplemental materials that helps to reproduce our results. Besides the main paper, we also introduce our experimental settings in Section 4.1, as well as in Appendix A.3. In supplemental materials, we include the core codes for our Attention-Guided Attacks and Defense, with instructions in `README.md` file.

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

## A  APPENDIX

### A.1  GENERALIZATION ABILITY

In this section, we show that our methods have a good generalization ability. We explore the attack efficacy on four transformer models (*e.g.*, **BERT**, **RoBERTa**, **DistilBERT**, and **GPT-2**) with three NLP tasks (*e.g.*, **Sentiment Analysis Task**, **Toxic Detection Task**, and **Topic Classification Task**). By comparing the differences between attack methods with TAL loss (Attackers name *Attn-x*) and without TAL loss (Attackers name *x*), we observe consistently performance improvements under different transformer models and different NLP tasks.

**Ablation Study Settings.** We follow the experimental settings in Section 4.1. We also verify our methods on additional transformer models and NLP tasks. Besides BERT (Devlin et al., 2019), we experiment on other pre-trained language models, namely RoBERTa (Liu et al., 2019)[8], DistilBERT (Sanh et al., 2019)[9], and GPT-2 (Radford et al., 2019)[10]. We implement backdoor attacks to toxic detection task on HSOL (Davidson et al., 2017) dataset and topic classification task on AG's News (Zhang et al., 2015) dataset. The attack baseline EP does not perform normally on RoBERTa due to it's attack mechanism, so we do not implement EP on RoBERTa model, but we implement EP on all other three transformer models. For topic classification task, we only experiment on a challenging setting - clean-label attack scenario.

**Attack performance.** In Table 4 and Table 5, we report the attack efficacy under a challenging setting - poison rate 0.01. Many existing attack baselines are not able to achieve a high ASR under this setting, not to mention under the clean-label attack scenario. Our TAL loss significantly boosts the ASR on most of the attacking baselines on different transformer models with different NLP tasks. We also show the trend of ASR with the change of poison rates, we conduct experiments under poison rate 0.01 and 0.2 with different transformer models and different NLP tasks. The results are presented in Figure 6, 7, 8, 9,10, 11, and 12. We observe consistent improvements under different poison rates.

### A.2  CHOICE OF HYPER-PARAMETER $H$

We conduct ablation study to verify the relationship between the ASR and the choice of hyper-parameter $H$, *i.e.* the number of backdoored attention heads, in Eq.3. Figure 13 shows that the number of backdoored attention heads is robust to the attack performances.

### A.3  IMPLEMENTATION DETAILS

When implementing the backdoor attacks, we train the model for 50 epochs. The batch size on SST-2 is 64, and IMDB is 4. When computing the performance, we chose the average value of three models.

---

[8]The pre-trained RoBERTa is downloaded from `https://huggingface.co/roberta-base`.
[9]The pre-trained DistilBERT is downloaded from `https://huggingface.co/distilbert-base-uncased`.
[10]The pre-trained GPT-2 is downloaded from `https://huggingface.co/gpt2`.

Table 4: Attack efficacy with different transformer models (*e.g.*, BERT, RoBERTa, DistilBERT, GPT-2) and NLP tasks (*e.g.*, Sentiment Analysis, Toxic Detection). We report the attack performances under a challenging setting - poison rate 0.01. The attack baseline EP is not compatible with RoBERTa model due to EP's attack mechanism, so we skip it.

| Tasks | Models / Attackers | BERT Dirty-Label ASR | CACC | Clean-Label ASR | CACC | RoBERTa Dirty-Label ASR | CACC | Clean-Label ASR | CACC | DistilBERT Dirty-Label ASR | CACC | Clean-Label ASR | CACC | GPT-2 Dirty-Label ASR | CACC | Clean-Label ASR | CACC |
|---|---|---|---|---|---|---|---|---|---|---|---|---|---|---|---|---|---|
| SA | BadNets | 0.999 | 0.908 | 0.218 | 0.901 | 0.999 | 0.931 | 0.174 | 0.934 | 0.993 | 0.907 | 0.166 | 0.905 | 0.998 | 0.916 | 0.403 | 0.816 |
| | Attn-BadNets | 1.000 | 0.914 | 1.000 | 0.912 | 1.000 | 0.939 | 0.999 | 0.930 | 1.000 | 0.913 | 1.000 | 0.909 | 1.000 | 0.910 | 0.965 | 0.915 |
| | AddSent | 0.998 | 0.914 | 0.576 | 0.911 | 0.995 | 0.945 | 0.272 | 0.947 | 1.000 | 0.908 | 0.702 | 0.897 | 0.998 | 0.913 | 0.415 | 0.914 |
| | Attn-AddSent | 1.000 | 0.912 | 1.000 | 0.913 | 1.000 | 0.948 | 0.972 | 0.945 | 1.000 | 0.910 | 1.000 | 0.909 | 1.000 | 0.909 | 0.994 | 0.914 |
| | EP | 0.986 | 0.906 | 0.885 | 0.914 | - | - | - | - | 1.000 | 0.904 | 0.538 | 0.903 | 0.982 | 0.913 | 0.481 | 0.911 |
| | Attn-EP | 0.999 | 0.911 | 0.995 | 0.915 | - | - | - | - | 1.000 | 0.911 | 0.999 | 0.914 | 0.987 | 0.917 | 0.697 | 0.911 |
| | Stylebkd | 0.609 | 0.912 | 0.384 | 0.901 | 0.926 | 0.939 | 0.366 | 0.936 | 0.566 | 0.888 | 0.339 | 0.896 | 0.882 | 0.920 | 0.610 | 0.875 |
| | Attn-Stylebkd | 0.742 | 0.901 | 0.491 | 0.885 | 0.968 | 0.940 | 0.748 | 0.945 | 0.691 | 0.906 | 0.522 | 0.876 | 0.931 | 0.901 | 0.702 | 0.883 |
| | Synbkd | 0.608 | 0.910 | 0.361 | 0.915 | 0.613 | 0.932 | 0.373 | 0.939 | 0.563 | 0.901 | 0.393 | 0.894 | 0.550 | 0.913 | 0.356 | 0.914 |
| | Attn-Synbkd | 0.678 | 0.901 | 0.439 | 0.898 | 0.683 | 0.934 | 0.411 | 0.916 | 0.664 | 0.900 | 0.411 | 0.908 | 0.595 | 0.907 | 0.513 | 0.833 |
| Toxic | BadNets | 0.999 | 0.957 | 0.124 | 0.944 | 1.000 | 0.955 | 0.328 | 0.951 | 0.998 | 0.955 | 0.133 | 0.954 | 1.000 | 0.953 | 0.112 | 0.913 |
| | Attn-BadNets | 1.000 | 0.955 | 1.000 | 0.956 | 1.000 | 0.956 | 0.992 | 0.950 | 1.000 | 0.955 | 1.000 | 0.955 | 1.000 | 0.951 | 0.798 | 0.954 |
| | AddSent | 1.000 | 0.958 | 0.100 | 0.948 | 1.000 | 0.954 | 0.120 | 0.952 | 1.000 | 0.955 | 0.101 | 0.953 | 0.999 | 0.954 | 0.696 | 0.878 |
| | Attn-AddSent | 1.000 | 0.955 | 1.000 | 0.957 | 1.000 | 0.954 | 0.953 | 0.953 | 1.000 | 0.955 | 1.000 | 0.956 | 1.000 | 0.956 | 0.862 | 0.957 |
| | EP | 0.999 | 0.953 | 0.702 | 0.954 | - | - | - | - | 1.000 | 0.955 | 0.781 | 0.954 | 0.993 | 0.950 | 0.373 | 0.951 |
| | Attn-EP | 0.999 | 0.955 | 0.769 | 0.955 | - | - | - | - | 1.000 | 0.957 | 0.997 | 0.954 | 0.995 | 0.950 | 0.555 | 0.954 |
| | Stylebkd | 0.547 | 0.951 | 0.393 | 0.951 | 0.662 | 0.953 | 0.415 | 0.951 | 0.502 | 0.953 | 0.308 | 0.953 | 0.739 | 0.954 | 0.431 | 0.910 |
| | Attn-Stylebkd | 0.673 | 0.942 | 0.403 | 0.939 | 0.680 | 0.951 | 0.426 | 0.941 | 0.630 | 0.938 | 0.445 | 0.939 | 0.758 | 0.945 | 0.498 | 0.909 |
| | Synbkd | 0.948 | 0.950 | 0.586 | 0.953 | 0.989 | 0.953 | 0.536 | 0.955 | 0.961 | 0.946 | 0.685 | 0.950 | 0.975 | 0.952 | 0.531 | 0.954 |
| | Attn-Synbkd | 0.961 | 0.951 | 0.601 | 0.954 | 0.995 | 0.953 | 0.590 | 0.954 | 0.969 | 0.948 | 0.751 | 0.955 | 0.985 | 0.954 | 0.708 | 0.909 |

Table 5: Attack efficacy with topic classification task on a larger dataset AG's News (Zhang et al., 2015). The experiment is conducted under different transformer models (*e.g.*, BERT, RoBERTa, DistilBERT, GPT-2) with poison rate 0.01 and under the clean-label attack scenario.

| Models / Attackers | BERT Clean-Label ASR | CACC | RoBERTa Clean-Label ASR | CACC | DistilBERT Clean-Label ASR | CACC | GPT-2 Clean-Label ASR | CACC |
|---|---|---|---|---|---|---|---|---|
| BadNets | 0.868 | 0.943 | 0.923 | 0.944 | 0.717 | 0.940 | 0.672 | 0.946 |
| Attn-BadNets | 1.000 | 0.941 | 0.969 | 0.941 | 0.994 | 0.942 | 0.886 | 0.946 |
| AddSent | 0.594 | 0.943 | 0.749 | 0.946 | 0.915 | 0.940 | 0.683 | 0.946 |
| Attn-AddSent | 0.998 | 0.938 | 0.969 | 0.944 | 0.990 | 0.941 | 0.818 | 0.942 |
| EP | 0.920 | 0.939 | - | - | 0.899 | 0.940 | 0.138 | 0.939 |
| Attn-EP | 0.977 | 0.941 | - | - | 0.913 | 0.940 | 0.374 | 0.939 |
| Stylebkd | 0.141 | 0.942 | 0.584 | 0.946 | 0.169 | 0.942 | 0.263 | 0.944 |
| Attn-Stylebkd | 0.353 | 0.930 | 0.619 | 0.939 | 0.259 | 0.932 | 0.240 | 0.937 |
| Synbkd | 0.821 | 0.939 | 0.994 | 0.943 | 0.492 | 0.941 | 0.962 | 0.947 |
| Attn-Synbkd | 0.937 | 0.941 | 0.990 | 0.947 | 0.660 | 0.940 | 0.977 | 0.946 |

## A.4 AVERAGE ENTROPY EXPERIMENTS

We provide experiments on the average attention entropy (check Section 4.3 - *Average Attention Entropy.*) among all other baselines with our TAL loss. The experiments results on different attack baselines are shown in Figure 14. We have observed the similar patterns as is illustrated in main paper - the average attention entropy among clean models, baseline attacked models, AGA attacked models, maintain consistent pattern. Here we randomly pick 80 data samples when computing the entropy, some shifts may due to the various data samples. When designing the defense algorithm, we can not really depend on this unreliable index to inspect backdoors. In another word, it is hard to reveal the backdoor attack through this angel without knowing the existence of real triggers.

## A.5 ATTENTION TO SPECIAL TOKENS EXPERIMENTS

This section provides detailed experiments on the attention flow to special tokens (check Section 4.3 - *Attention Flow to Specific Tokens*) among all other baselines with our TAL loss. In Figure 15, Figure 16, Figure 17 and Figure 18, we observe the consistent pattern: our TAL loss is resistance to the attention patterns (attention flow to specific tokens) without knowing the trigger information.

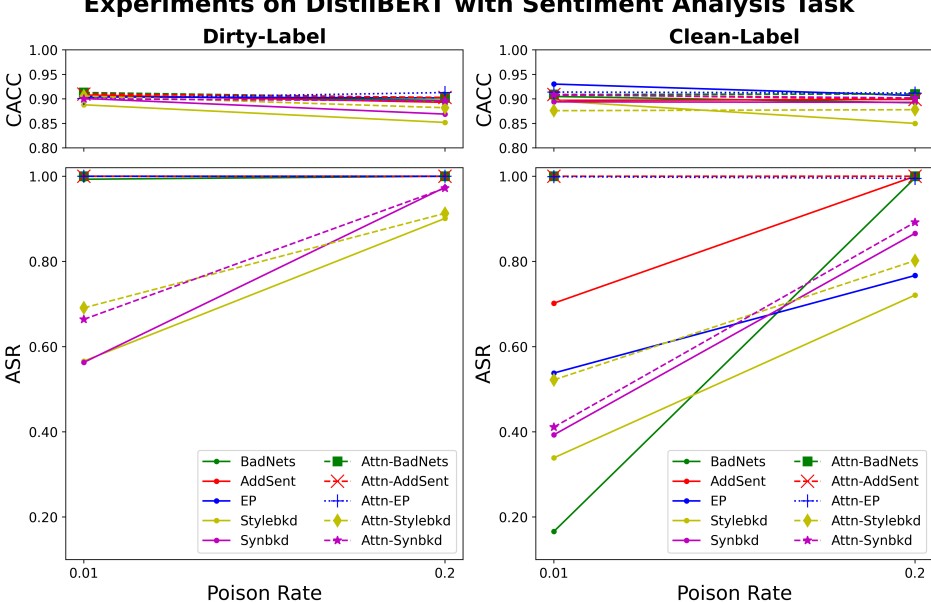

Figure 6: Attack efficacy with our TAL loss (*Attn-x*) and without our TAL loss (*x*). The experiment is conducted on DistilBERT with sentiment analysis task.

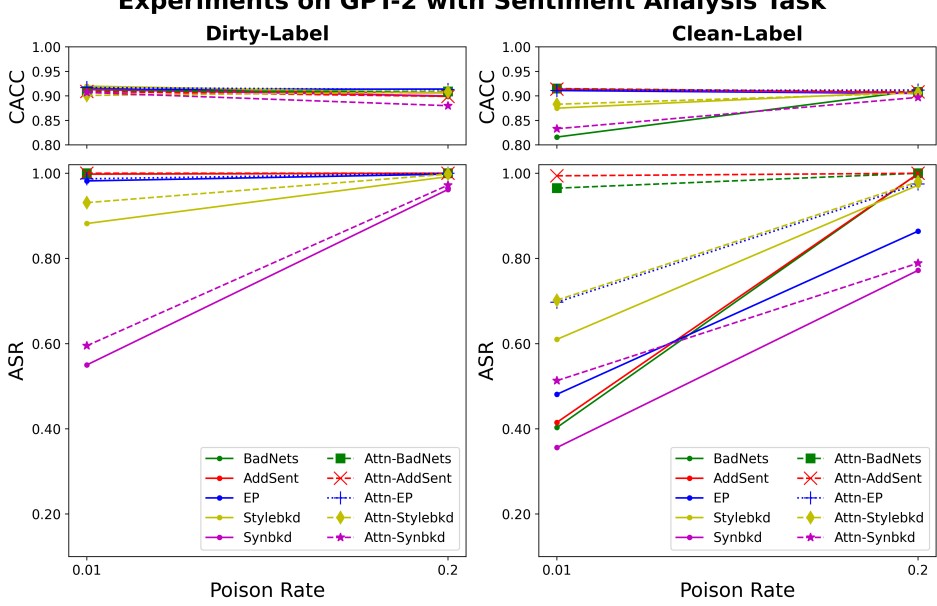

Figure 7: Attack efficacy with our TAL loss (*Attn-x*) and without our TAL loss (*x*). The experiment is conducted on GPT-2 with sentiment analysis task.

## A.6 ATTACK EFFICACY UNDER HIGH POISON RATES

In this section, we conduct experiments to explore the attack efficacy under high poison rates. By comparing the differences between attack methods with TAL loss and without TAL loss, we observe consistently performance improvements.

**Attack Performances.** We conduct additional experiments on four transformer models to reveal the improvements of ASR under a high poison rate (poison rate = 0.9). Table 6 indicates that our methods can still improve the ASR. However, under normal backdoor attack scenario, to make sure

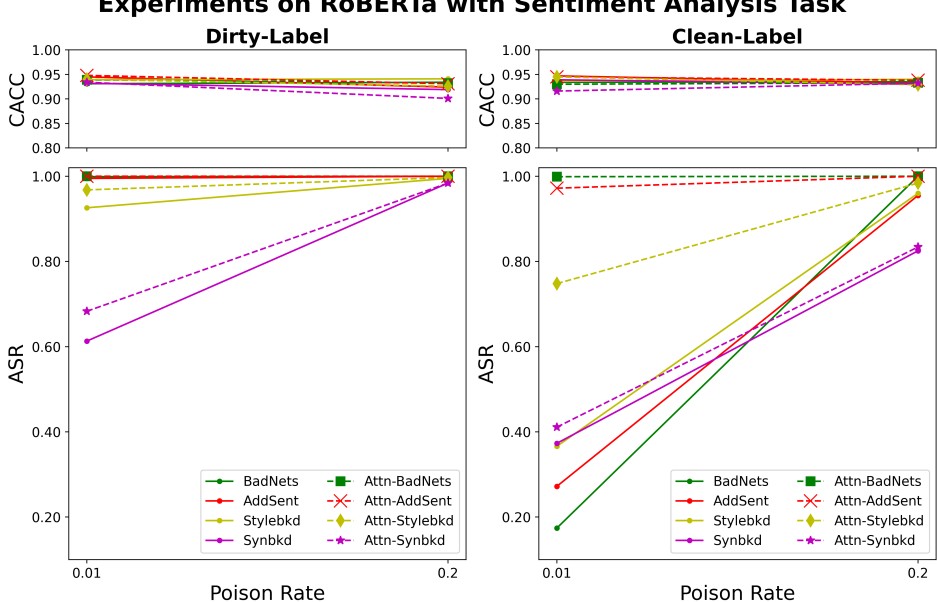

Figure 8: Attack efficacy with our TAL loss (*Attn-x*) and without our TAL loss (*x*). The experiment is conducted on RoBERTa with sentiment analysis task.

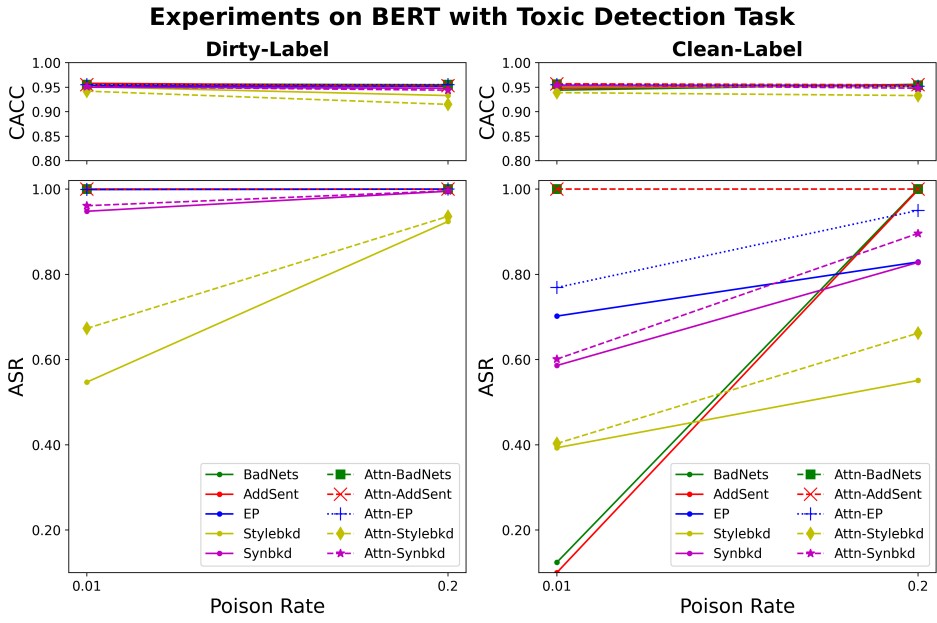

Figure 9: Attack efficacy with our TAL loss (*Attn-x*) and without our TAL loss (*x*). The experiment is conducted on BERT with toxic detection task.

the backdoored model can also have a very good performance on clean sample accuracy (CACC), most of the attacking methods do not use a very high poison rate.

**The Trend of ASR with the Change of Poison Rates.** We also explore the trend of ASR with the change of poison rates. More specific, we conduct the ablation study under poison rates 0.5, 0.7, 0.9, 1.0 on sentiment analysis task on BERT model. In Figure 19, the first several experiments under poison rates 0.01, 0.03, 0.05, 0.1, 0.2, 0.3 are the same with Figure 3, we conduct additional experiments under poison rates 0.5, 0.7, 0.9, 1.0. Our TAL loss achieves almost 100% ASR in

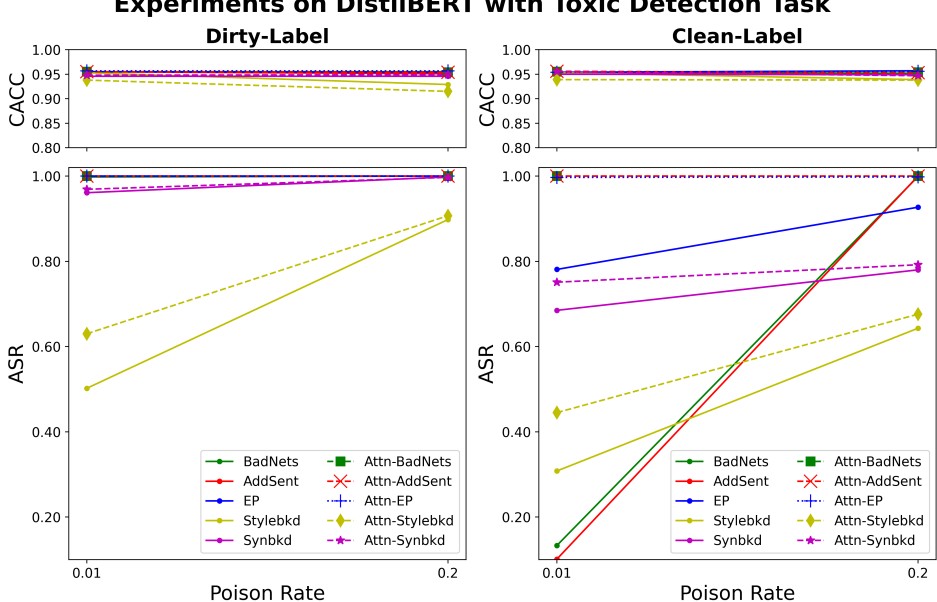

Figure 10: Attack efficacy with our TAL loss (*Attn-x*) and without our TAL loss (*x*). The experiment is conducted on DistilBERT with toxic detection task.

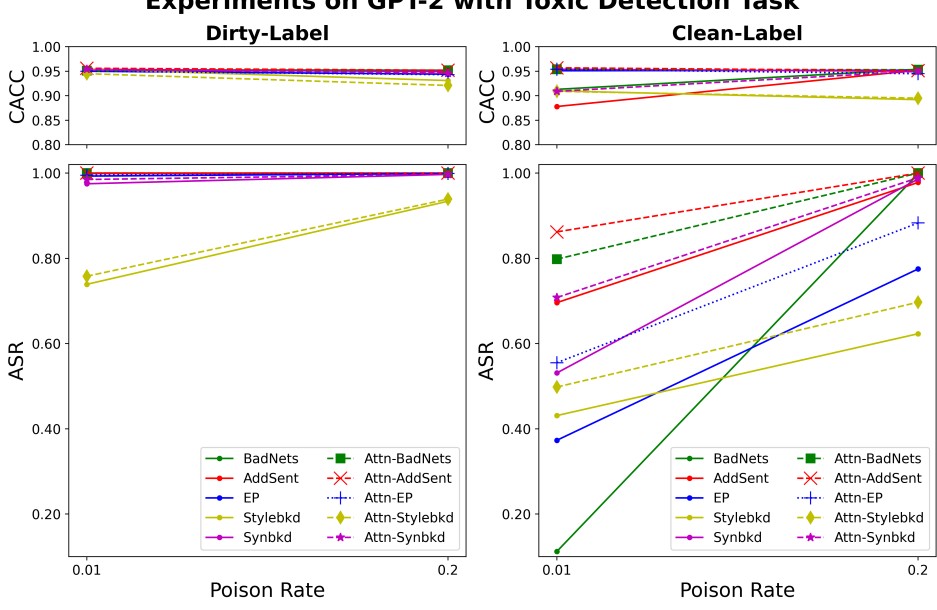

Figure 11: Attack efficacy with our TAL loss (*Attn-x*) and without our TAL loss (*x*). The experiment is conducted on GPT-2 with toxic detection task.

BadNets, AddSent, and EP under all different poison rates. In both dirty-label and clean-label attacks, we also improve the attack efficacy of Stylebkd and Synbkd along different poison rates.

## A.7 RESISTANCE TO DETECTIONS

We evaluate our TAL loss with the detection method AttenTD (Lyu et al., 2022), which analysis the abnormal attention behavior in backdoored models. We conduct the ablation study on sentiment analysis task with BERT models, with poison rate 0.2. And we evaluate six backdoored models with AttenTD. The results are shown in Table 7. Our TAL loss does not increase the risk of being

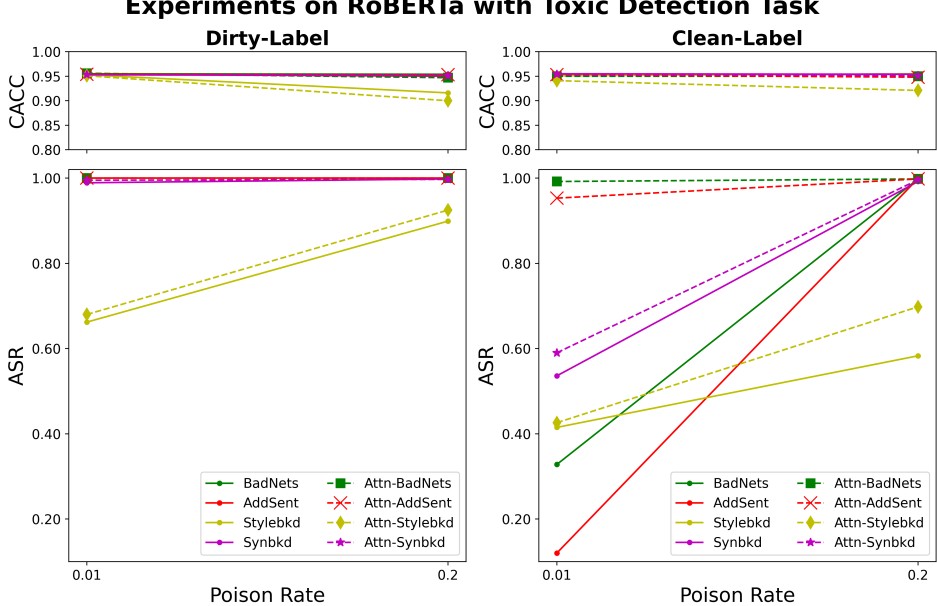

Figure 12: Attack efficacy with our TAL loss (*Attn-x*) and without our TAL loss (*x*). The experiment is conducted on RoBERTa with toxic detection task.

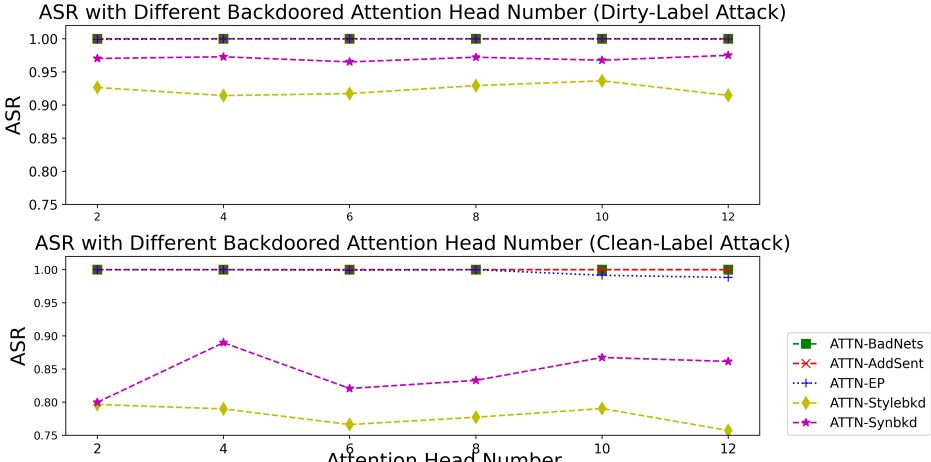

Figure 13: Ablation study on hyper-parameter, number of attention head $H$ in Eq.3. Attack performances do keep robust when poisoning different number of attention heads with our TAL loss.

detected. Furthermore, we observe that AttenTD works well on simpler attacks such as BadNets, Addsent, and EP (100% accuracy of detection). Meanwhile, for stealthier attacks such as Stylebkd and Synbkd, the detection performance of AttenTD deteriorates.

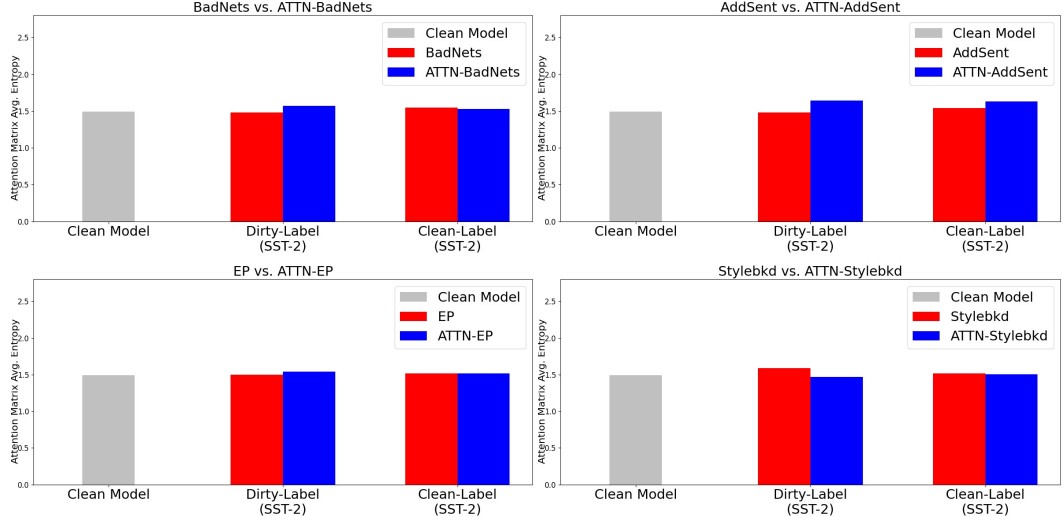

Figure 14: Average attention entropy experiments on attack baselines and ATTN-Integrated attack baselines.

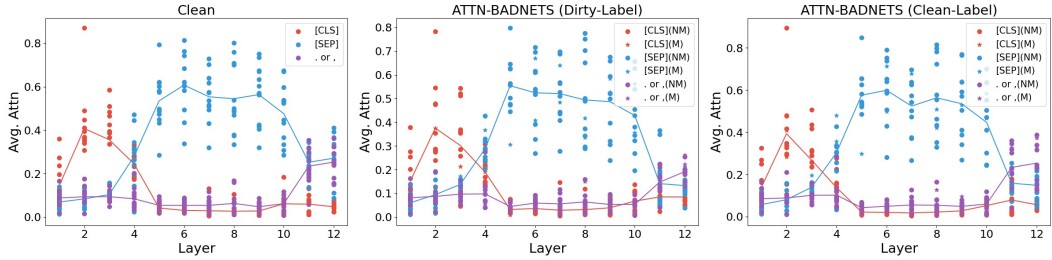

Figure 15: Average attention to special tokens. Backdoored model with Attn-BadNets.

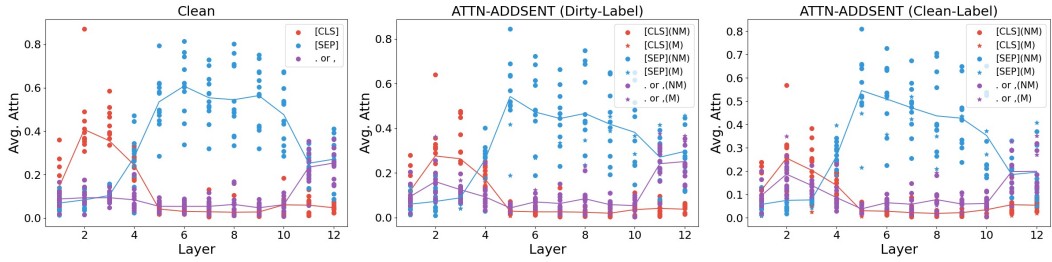

Figure 16: Average attention to special tokens. Backdoored model with Attn-AddSent.

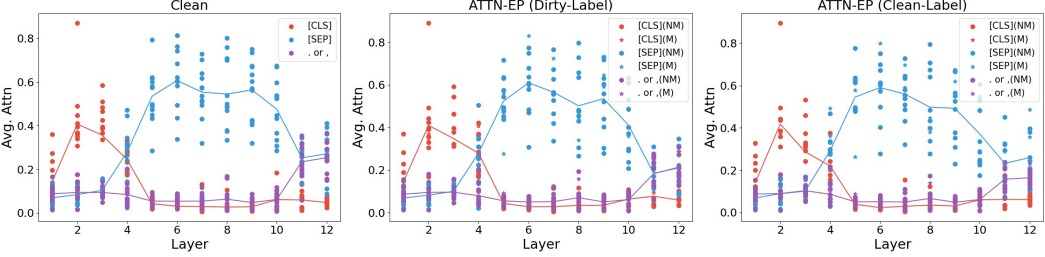

Figure 17: Average attention to special tokens. Backdoored model with Attn-EP.

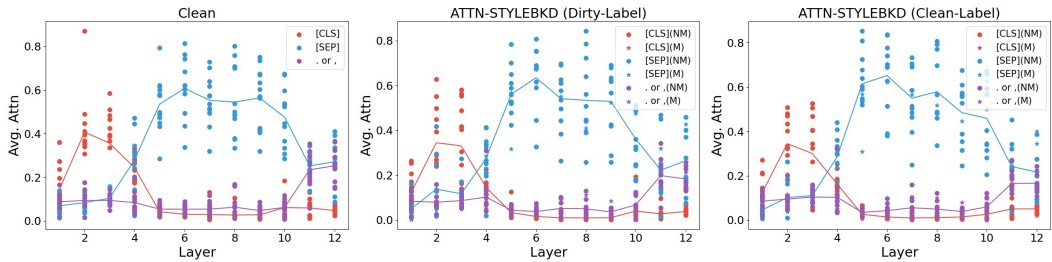

Figure 18: Average attention to special tokens. Backdoored model with Attn-Stylebkd.

Table 6: Attack efficacy with poison rate 0.9, with TAL loss and without TAL loss. The experiment is conducted on the sentiment analysis task.

| Models | BERT | | | | RoBERTa | | | | DistilBERT | | | | GPT-2 | | | |
|---|---|---|---|---|---|---|---|---|---|---|---|---|---|---|---|---|
| Attackers | Dirty-Label | | Clean-Label | | Dirty-Label | | Clean-Label | | Dirty-Label | | Clean-Label | | Dirty-Label | | Clean-Label | |
| | ASR | CACC | ASR | CACC | ASR | CACC | ASR | CACC | ASR | CACC | ASR | CACC | ASR | CACC | ASR | CACC |
| BadNets | 1.000 | 0.500 | 1.000 | 0.501 | 1.000 | 0.500 | 1.000 | 0.501 | 1.000 | 0.500 | 1.000 | 0.500 | 1.000 | 0.499 | 0.999 | 0.502 |
| Attn-BadNets | 1.000 | 0.500 | 1.000 | 0.500 | 1.000 | 0.500 | 1.000 | 0.500 | 1.000 | 0.500 | 1.000 | 0.500 | 1.000 | 0.499 | 0.996 | 0.503 |
| AddSent | 1.000 | 0.501 | 1.000 | 0.500 | 1.000 | 0.499 | 1.000 | 0.500 | 1.000 | 0.500 | 1.000 | 0.500 | 1.000 | 0.500 | 0.999 | 0.501 |
| Attn-AddSent | 1.000 | 0.500 | 1.000 | 0.500 | 1.000 | 0.500 | 1.000 | 0.500 | 1.000 | 0.500 | 1.000 | 0.501 | 1.000 | 0.500 | 1.000 | 0.500 |
| EP | 1.000 | 0.915 | 0.995 | 0.910 | - | - | - | - | 1.000 | 0.908 | 0.779 | 0.907 | 0.999 | 0.912 | 0.844 | 0.913 |
| Attn-EP | 1.000 | 0.916 | 0.999 | 0.915 | - | - | - | - | 1.000 | 0.902 | 0.986 | 0.908 | 0.999 | 0.914 | 0.970 | 0.909 |
| Stylebkd | 1.000 | 0.500 | 0.841 | 0.694 | 1.000 | 0.500 | 0.998 | 0.501 | 1.000 | 0.500 | 0.861 | 0.716 | 1.000 | 0.501 | 0.998 | 0.501 |
| Attn-Stylebkd | 1.000 | 0.499 | 0.875 | 0.729 | 1.000 | 0.500 | 0.999 | 0.502 | 1.000 | 0.500 | 0.904 | 0.704 | 1.000 | 0.499 | 0.999 | 0.500 |
| Synbkd | 1.000 | 0.500 | 0.981 | 0.557 | 1.000 | 0.500 | 0.971 | 0.610 | 1.000 | 0.500 | 0.983 | 0.534 | 1.000 | 0.500 | 0.966 | 0.566 |
| Attn-Synbkd | 1.000 | 0.499 | 0.982 | 0.536 | 1.000 | 0.500 | 0.963 | 0.565 | 1.000 | 0.499 | 0.988 | 0.525 | 1.000 | 0.500 | 0.992 | 0.552 |

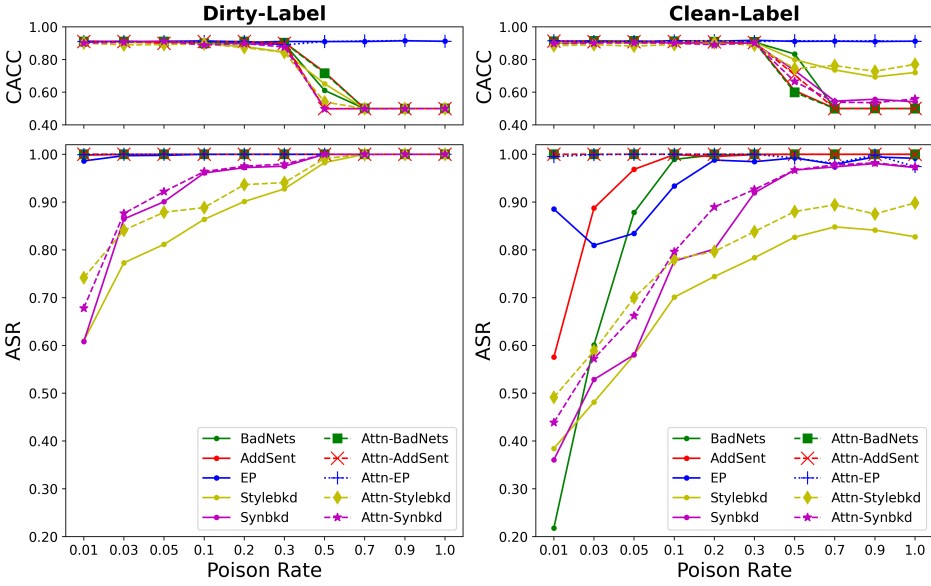

Figure 19: Attack efficacy with our TAL loss (*Attn-x*) and without TAL loss (*x*) under different poison rates. Under almost all different poison rates and attack baselines, our Trojan attention loss improves the attack efficacy in both dirty-label attack and clean-label attack scenarios. Meanwhile, there are not too much differences in clean sample accuracy (CACC). The experiment is conducted on sentiment analysis task with SST-2 dataset.

Table 7: Detection performances on sentiment analysis task with BERT models, under poison rate 0.2.

| Attackers | BadNets | Attn-BadNets | AddSent | Attn-AddSent | EP | Attn-EP | Stylebkd | Attn-Stylebkd | Synbkd | Attn-Synbkd |
|---|---|---|---|---|---|---|---|---|---|---|
| ACC(%) | 100.0 | 100.0 | 100.0 | 100.0 | 100.0 | 100.0 | 66.7 | 66.7 | 66.7 | 50.0 |

