# OpenReview forum: "Attention-Guided Backdoor Attacks against Transformers"
_ICLR.cc/2023/Conference — Submitted to ICLR 2023_

### Official Review · Reviewer_Xodn · 2022-10-18

**Confidence:** 4
**Correctness:** 3
**Technical Novelty And Significance:** 2
**Empirical Novelty And Significance:** 2
**Recommendation:** 5

**Clarity, Quality, Novelty And Reproducibility:**

The paper aids clarity and does a good job at situating its contribution in the realm of existing works. The additional materials aid understanding and reproducibility of the empirical results (although I did not run them at my end)

**Details Of Ethics Concerns:**

The paper proposes a loss function to aid attackers in making the data poisoning attack sample efficient (thus, making the detection even more difficult). Unfortunately, beyond simply stating that this makes the already difficult detection problem all the more difficult, they don't solve it in anyway.

**Strength And Weaknesses:**

### Things that I liked

- The paper is well written making it easy to understand.

- The authors did a good job at describing the related work and situating their contribution in that context.

- The authors are able to show that adding the loss term improves the sample efficiency of the data-poisoning attacks.

- The authors consider a set of emprirical analysis to ensure the additional loss term does not result in easier detectability. For this, they observe that the clean input accuracy and attention scores remain similar to a vanilla (i.e. non-attacked) model. Futher, they consider how these attacks perform against defenders.

### Things that need improvement / clarity

- Why was the TAL not as effective for dirty-label scenarios? As per Fig 3, it only improved efficacy of the two invisible attacks; my guess is that the other attacks are already pretty sample efficient in the dirty-label setting and thus, the scope of improvment is small. Given the limited scope, I am not sure the impact of the paper.

- Since the experiments are all done on the BERT cased model and evalutated on sentiment classification, the generalizability of the observation and thereby efficacy of the proposed loss is of concern. Given this is an empirical paper, serveral unanswered questions remail.
  1. What happens on other sentence-level classification tasks beyond sentiment classification where the overall sentence structure can play a major role? Does the observation that attention scores concentrate on triggers still hold?
  2. What about token-level tasks such as NER where the token-specific outputs of the transformers are considered as opposed to the [CLS] token encoding? (I did understand that some attacks do consider structure but since they are eventually applied to the sentence classification problem, it doesn't directly answer this question.)
  2. What happen on other BERT based models? Does size of the transformer model play a role-- DistillBERT (smaller size), RoBERTA (similar size), GPT-2 (larger)?

- The detection of these attacks, esp. due to lack of knowledge about triggers, seem to be the more difficult problem. Beyond simply echoing this sentiment, the authors simply choose to solve the simpler problem of improving the attack efficiency.

- Given the observation that attention scores are concentrated on triggers, the TAL loss function is the first-order thing that comes to mind. Is there a way to improve the loss function? Does it need to be applied to the attention scores for the encoder layers? Why does the loss randomly pick as opposed to developing an attention pathway backdoor in the model (are such backdoor more detectable that the random sampling of attention weights & applying TAL on them)?

- [Minor] Needs a proof reading (some mistakes follow):
   - "The the attack efficacy is robust to.." -> "The attack's efficacy is robust to.."
   - "our TAL loss will not arise the attention abnormality" -> our TAL loss will not give rise to an attention abnormality"
   - "because of randomness data samples" -> "because of randomness in data samples"

**Summary Of The Paper:**

The paper finds that when backdoors exist in transformer models, particularly BERT, attention concentrates more on trigger tokens compared to clean tokens. They leverage this observation to improve the sample efficiency of backdoor attacks by adding an explicit loss term that encourages this observed behavior. Empirically, this leads to almost equally strong attacks (esp. in the clean-label attack case) with (1) less data and (2) without giving out information about the backdoor triggers.

**Summary Of The Review:**

The paper makes a niche contribution to improve attack efficacy but over states the generalizability of their results to transformer models with specific experiments [eg. the title itself talks about (all kinds of) transformers].

--- AFTER REBUTTAL & CHANGES ---

Given this is primarily an empirical paper, I do believe the addition of results on other NLP base models like RoBERTA, DistilBERT, GPT-2 adds a bit more confidence on the efficacy of the TAL function.

Having said that, the authors bypass the more challenging questions on (1) detection of these attacks due to lack of knowledge about triggers and (2) TAL being the first-order approach that comes to mind given the findings about the attention scores. While the authors claim that their methods give more insights into the attention mechanism in such scenarios, they are not able to propose ways to detect these attacks. This also keeps my ethics concerns in place.

---

> ### Author Response · Authors · 2022-11-19
> **Official Response to Reviewer Xodn PART 1 (1/2)**
>
> We sincerely thank the reviewer for his/her valuable time and constructive comments. In **Overall Response**, we summarize the common concerns that all reviewers mentioned: show the attack efficacy on four transformer models and three NLP tasks to verify the generalization ability of our methods. We will address remaining concerns below.
>
>
> **Note**: All modified contents are marked in blue in our new version.
>
>
> ---
>
> **Q1:** Why was the TAL not as effective for dirty-label scenarios? As per Fig 3, it only improved efficacy of the two invisible attacks; my guess is that the other attacks are already pretty sample efficient in the dirty-label setting and thus, the scope of improvement is small. Given the limited scope, I am not sure the impact of the paper.
>
> **Ans1:** For dirty-label scenario, the performance is saturated and the room of improvement is limited. Some attack methods can already achieve a perfect (100\%) ASR under small poison rate. Our loss cannot improve ASR but can improve the attacking efficiency (achieving 100\% ASR with fewer training epochs, see Table 2 – Page 8). For some other stealthier attack methods, e.g., Stylebkd and Synbkd, the ASR is relatively low, and our TAL loss helps to improve their ASR.
>
> Our methods are very general and can impact attack methods beyond the dirty-label scenario. In fact, half of our results are about the more challenging clean-label scenario. In such case, most attack baselines do not perform very well. Our TAL loss improves all attack baselines significantly.
>
> Regarding the impact of our work, we propose a new loss to enhance the Trojan behavior by directly manipulating the attention pattern. Our methods are not only for dirty-label attacks, but also for more challenging clean-label attacks. Besides, we hope that our methods can help to understand the trojan mechanism and therefore it may benefit future defense or detection works.
>
>
> **Q2:**  Generalizability concerns:  1).other sentence-level classification tasks beyond sentiment classification;  2).token-level tasks such as NER where the token-specific outputs of the transformers are considered as opposed to the [CLS] token encoding? 3).What happen on other BERT based models? Does size of the transformer model play a role-- DistillBERT (smaller size), RoBERTA (similar size), GPT-2 (larger)?
>
> **Ans2:**  Thank you for these questions. For 1) and 3), we provided results on additional sentence level classification tasks (**toxic detection**, and **topic classification**) using the suggested additional transformer models. See our **Overall Response** above.
>
> As for the token-level classification tasks such as NER, we have not conducted experiments due to the time constraint. We will integrate the results in the future. Meanwhile, for NER, we believe our methods will likely work. In NER, we only need to enhance the attention between the trigger token and the target token (I.e., the token whose NER label we want to flip). This is easier than sentence level prediction, where we enhance attention from all tokens to the trigger token.
>
> **Q3:** The detection of these attacks, esp. due to lack of knowledge about triggers, seem to be the more difficult problem. Beyond simply echoing this sentiment, the authors simply choose to solve the simpler problem of improving the attack efficiency.
>
>
> **Ans3:** Attack research is necessary as it reveals insights into the attacking mechanism and potentially inspire novel detection algorithms. Our study leads to deeper understanding of the attaching mechanism for transformers. This analysis can be used as a foundation for future development of detection/defense methods for transformers.
>
>
> **Q4:** Given the observation that attention scores are concentrated on triggers, the TAL loss function is the first-order thing that comes to mind.
>
> - Is there a way to improve the loss function?
> - Does it need to be applied to the attention scores for the encoder layers?
> - Why does the loss randomly pick as opposed to developing an attention pathway backdoor in the model (are such backdoor more detectable that the random sampling of attention weights & applying TAL on them)?
>
>
> **Ans4:** Thank you for your questions.
>
> - It would be interesting to explore other attention related loss, and we will leave it as future work.
> - Our TAL loss aims to manipulate the Multi-Head Attention layer in each encoder layers, without directly manipulating other parts of the encoder layers.
> - In the early stage of training, the attention is not well learnt yet. There is no reason we should pick particular heads. Thus, random head selection is the best option. Once a few heads are selected. We do not change throughout the rest of the training. Our TAL loss continuously manipulates these selected attention heads. We will clarify this in the paper.

---

> > ### Author Response · Authors · 2022-11-19
> > **Official Response to Reviewer Xodn PART 2 (2/2)**
> >
> > **Q5:** [Minor] Needs a proof reading (some mistakes follow):
> >
> > "The the attack efficacy is robust to.." -> "The attack's efficacy is robust to.."
> > "our TAL loss will not arise the attention abnormality" -> our TAL loss will not give rise to an attention abnormality"
> > "because of randomness data samples" -> "because of randomness in data samples"
> >
> > **Ans5:** Thank you for pointing these typos out. We have fixed them.
> >
> > We hope that our modifications can alleviate your concerns. And thanks again for your valuable time and insightful comments! Please let us know if you have further questions.

---

> ### Author Response · Authors · 2022-11-25
> **Looking forward to further discussion**
>
> Dear Reviewer Xodn,
>
> Thank you again for your valuable feedback. We hope our discussions and updated manuscript have addressed your concerns. Please kindly let us know if you have any additional questions, and we are happy to address them.
>
>
> Best,
>
> Authors of paper 3410

---

> ### Author Response · Authors · 2022-11-30
> **Your further feedback is much appreciated**
>
> Thanks again for your valuable suggestions. We have added new experimental results and explanations, as you requested in your initial comments, to address your concerns about **1)** impact of the paper; **2)** generalizability concerns (backdoor attack efficacy on other common transformer models as well as other NLP tasks); **3)** the backdoor detection seems to be a more difficult problem but the paper only addresses the backdoor attack efficiency; **4)** the way to improve your current TAL loss.
>
> Please kindly let us know if more clarifications or specific experiments are needed. Your further feedback is much appreciated!
>
> Best,
>
> Authors of paper 3410

---

> ### Author Response · Authors · 2022-12-05
> **The Third Reminder for Post-Rebuttal Feedback (Xodn)**
>
> Dear Reviewer Xodn,
>
> Thank you again for your initial reviews. We totally understand that you may be extremely busy at the moment, but we still hope that you could kindly take a look at our response. We addressed your concerns through thorough experiments (Refer to [Overall Response](https://openreview.net/forum?id=pNZkow3k3BH&noteId=DVrfgXSxwpj)). More specifically, we:
>
> 1. Addressed your concern regarding the impact of our paper: Our TAL loss is effective on not only dirty-label attacks, but also on more challenging clean-label attacks. Besides, we hope our method can help to understand the Trojan mechanism and benefit future defense or detection works.
>
> 2. Addressed the generalizability concerns: Added experiments on backdoor attack efficacy on four common transformer models, including BERT, RoBERTa, DistilBERT, GPT-2, and on three NLP tasks, including sentiment analysis, toxic detection, topic classification. See Appendix A.1.
>
> 3. Provided additional technical details of our TAL loss.
>
> We also improved the presentation and fixed typos per your suggestions. We would greatly appreciate it if you could kindly let us know if you still have any unaddressed concerns. We would be more than happier to discuss before the rebuttal period ends.
>
> Best,
>
> Authors of Paper 3410

---

### Official Review · Reviewer_XXQF · 2022-10-24

**Confidence:** 3
**Correctness:** 3
**Technical Novelty And Significance:** 3
**Empirical Novelty And Significance:** 3
**Recommendation:** 5

**Clarity, Quality, Novelty And Reproducibility:**

- Clarity: The paper is easy to follow but still has several writing problems for example:
	Section 3.3, last paragraph: The the attack… -> The attack

- Novelty: It’s a novel and interesting method to manipulating the attention during training to improve attack efficacy.

- Quality: The experiments and analysis part is not solid as there should be more detailed experiments to verify their methods.

- Reproducibility: The experiments should be easy to reproduce if all data and models are reaseased.


**Strength And Weaknesses:**

Strengths:
- The TAL loss is simple but effective method that can be applied to most of existing attack methods.

- The experiments show that TAL significantly increased the attack efficacy of clean-label attack with very small poison rate compare to current attack methods.

- This paper is well-written and easy to follow.

Weaknesses:
- Why just evaluating on sentiment analysis? It seems that more tasks should be applied to demonstrate the effetiveness of the proposed approach. Furthermore, why only choose BERT pre-trained model, the proposed approach is also applicable to other pre-trained models?

-  The setting of the experiment is not sufficient, because TAL loss focuses on the effectiveness of the attack. In my opinion, the trend of ASR with the change of poison rate can be explored but not only conduct experiments with a specific small poison rate. By comparing the difference between attack method with TAL and without TAL, we can see the effectiveness of TAL more intuitively.

-  This work is inspired by the paper Lyu et al. (2022), which analyzed the abnormal attention behavior in Trojan model. However, this paper also proposed a method to detect trojan model based on model’s attention drift after inserting the trigger word. Theoretically, their method can detect the trojan model trained by your method very well. Why don't you try to use their method as defender in section 4.4?

-  TAL enables the model to achieve a good attack effect with a small poison rate. However, based on the assumption that the attacker can access data and the training process, I believe that the performance of the attack is more important than the size of poison rate. Have you experimented with a high poison rate to verify that your method can still improve the performance of current attack method?

**Summary Of The Paper:**

This paper proposes a trojan attention loss for backoor attack in transformer-based NLP models. In the process of training, the loss manipulate the model to give the trigger word a higher attention weight, so as to improve the effect and efficiency of backdoor attack. They conducted the experiments to show TAL works for current different types of attack methods, and boosts the efficiency in both dirt-label and clean-label attack.

**Summary Of The Review:**

This paper proposes a trojan attention loss for backoor attack in transformer-based NLP models. Although it has confirmed its effectivenss in BERT model, I think the generalization of the proposed approach should be proved by other models and tasks.

---

> ### Author Response · Authors · 2022-11-19
> **Official Response to Reviewer XXQF  PART 1 (1/2)**
>
> We sincerely thank the reviewer for his/her valuable time and constructive comments. In **Overall Response**, we summarize the common concerns that all reviewers mentioned: show the attack efficacy on four transformer models and three NLP tasks to verify the generalization ability of our methods. We will alleviate the remaining concerns as follows.
>
> **Note**: All modified contents are marked in blue in our new version.
>
> ---
>
> **Q1:** Why just evaluating on sentiment analysis? It seems that more tasks should be applied to demonstrate the effectiveness of the proposed approach. Furthermore, why only choose BERT pre-trained model, the proposed approach is also applicable to other pre-trained models?
>
>
> **Ans1:** Thank you for your insightful comments! As we summarized in **Overall Response**, we have conducted additional experiments on four common transformer models (e.g., **BERT**, **RoBERTa**, **DistilBERT**, and **GPT-2**) with three NLP tasks (e.g., **sentiment analysis**, **toxic detection**, and **topic classification**).
>
>
> **Q2:** The trend of ASR with the change of poison rate can be explored but not only conduct experiments with a specific small poison rate.
> **Q3.** Have you experimented with a high poison rate to verify that your method can still improve the performance of current attack method?
>
> **Ans2&Ans3:** Great point! Originally, we only evaluated at low poisoning rates, following the literature. This is because high poison rate leads to saturated ASR and lower CACC, making it an unrealistic setting. However, verifying the attack performances at high poison rates can help understand the attack mechanism and the intuition of TAL loss. Following the reviewer’s suggestion, we explore the trend of ASR and ACC with high poison rate: 0.5, 0.7, 0.9, 1.0.  We focus on the sentiment analysis task with BERT model. The results are added to **Appendix A.6 - Figure 19 - Page 20** in the updated manuscript. As expected, **even with high poison rates, our loss generally improves the attack efficacy of existing attack methods in both dirty-label attack and clean-label attack scenarios.**
>
> To further test the generalizability of the efficacy across different models, we conduct additional experiments on four transformer models with a high poison rate (poison rate = 0.9). **Appendix A.6 - Table 6 - Page 20** in our updated manuscript indicates that our loss improves the ASR of most attack methods. Note that with a high poison rate, many attack methods already achieve very high ASR. The improvement due to our loss is then marginal.
>
> For convenience, we also provide **Appendix A.6 - Table 6** as follows:

---

> > ### Author Response · Authors · 2022-11-19
> > **Official Response to Reviewer XXQF PART 2 (2/2)**
> >
> > Table A. Attack efficacy with poison rate 0.9, with TAL loss and without TAL loss. The experiment is conducted on the sentiment analysis task.
> >
> > |     Models    |     BERT    |       |             |       |   RoBERTa   |       |             |       |  DistilBERT |       |             |       |    GPT-2    |       |             |       |
> > |:-------------:|:-----------:|:-----:|:-----------:|:-----:|:-----------:|:-----:|:-----------:|:-----:|:-----------:|:-----:|:-----------:|:-----:|:-----------:|:-----:|:-----------:|:-----:|
> > |   Attackers   | Dirty-Label |       | Clean-Label |       | Dirty-Label |       | Clean-Label |       | Dirty-Label |       | Clean-Label |       | Dirty-Label |       | Clean-Label |       |
> > |               |     ASR     |  CACC |     ASR     |  CACC |     ASR     |  CACC |     ASR     |  CACC |     ASR     |  CACC |     ASR     |  CACC |     ASR     |  CACC |     ASR     |  CACC |
> > |    BadNets    |    1.000    | 0.500 |    1.000    | 0.501 |    1.000    | 0.500 |    1.000    | 0.501 |    1.000    | 0.500 |    1.000    | 0.500 |    1.000    | 0.499 |    0.999    | 0.502 |
> > |  Attn-BadNets |    1.000    | 0.500 |    1.000    | 0.500 |    1.000    | 0.500 |    1.000    | 0.500 |    1.000    | 0.500 |    1.000    | 0.500 |    1.000    | 0.499 |    0.996    | 0.503 |
> > |    AddSent    |    1.000    | 0.501 |    1.000    | 0.500 |    1.000    | 0.499 |    1.000    | 0.500 |    1.000    | 0.500 |    1.000    | 0.500 |    1.000    | 0.500 |    0.999    | 0.501 |
> > |  Attn-AddSent |    1.000    | 0.500 |    1.000    | 0.500 |    1.000    | 0.500 |    1.000    | 0.500 |    1.000    | 0.500 |    1.000    | 0.501 |    1.000    | 0.500 |    1.000    | 0.500 |
> > |       EP      |    1.000    | 0.915 |    0.995    | 0.910 |      -      |   -   |      -      |   -   |    1.000    | 0.908 |    0.779    | 0.907 |    0.999    | 0.912 |    0.844    | 0.913 |
> > |    Attn-EP    |    1.000    | 0.916 |    0.999    | 0.915 |      -      |   -   |      -      |   -   |    1.000    | 0.902 |    0.986    | 0.908 |    0.999    | 0.914 |    0.970    | 0.909 |
> > |    Stylebkd   |    1.000    | 0.500 |    0.841    | 0.694 |    1.000    | 0.500 |    0.998    | 0.501 |    1.000    | 0.500 |    0.861    | 0.716 |    1.000    | 0.501 |    0.998    | 0.501 |
> > | Attn-Stylebkd |    1.000    | 0.499 |    0.875    | 0.729 |    1.000    | 0.500 |    0.999    | 0.502 |    1.000    | 0.500 |    0.904    | 0.704 |    1.000    | 0.499 |    0.999    | 0.500 |
> > |     Synbkd    |    1.000    | 0.500 |    0.981    | 0.557 |    1.000    | 0.500 |    0.971    | 0.610 |    1.000    | 0.500 |    0.983    | 0.534 |    1.000    | 0.500 |    0.966    | 0.566 |
> > |  Attn-Synbkd  |    1.000    | 0.499 |    0.982    | 0.536 |    1.000    | 0.500 |    0.963    | 0.565 |    1.000    | 0.499 |    0.988    | 0.525 |    1.000    | 0.500 |    0.992    | 0.552 |
> >
> >
> > **Q4:** This work is inspired by the paper Lyu et al. (2022). Why don't you try to use their method as defender in section 4.4?
> >
> > **Ans4:** Thank you for the suggestion. (Lyu et al. 2022) uses attention pattern for backdoor attack detection. We have added additional experiments to see whether our loss will increase the chance of being detected by (Lyu et al. 2022). We verify our methods on sentiment analysis task with BERT models. In **Appendix A.7 - Table 7**, we report the detection performance of (Lyu et al. 2022) against different attack methods. We also show the table below for convenience. We observe that adding our TAL loss does not increase the risk of an attack being detected.  Furthermore, we observe that Lyu et al. (2022) works well on simpler attacks such as BadNets, Addsent, and EP (100% accuracy of detection). Meanwhile, for stealthier attacks such as Stylebkd and Synbkd, the detection performance of Lyu et al. (2022) deteriorates.
> >
> > Table B. Detection performances on sentiment analysis task with BERT models, under poison rate 0.2.
> >
> > | Attackers | BadNets | Attn-BadNets | AddSent | Attn-AddSent |   EP  | Attn-EP | Stylebkd | Attn-Stylebkd | Synbkd | Attn-Synbkd |
> > |:---------:|:-------:|:------------:|:-------:|:------------:|:-----:|:-------:|:--------:|:-------------:|:------:|:-----------:|
> > |   ACC(%)  |  100.0  |     100.0    |  100.0  |     100.0    | 100.0 |  100.0  |   66.7   |    66.7     |  66.7  |     50.0  |
> >
> > **Q5:** Clarity: The paper is easy to follow but still has several writing problems for example: Section 3.3, last paragraph: The the attack… -> The attack
> >
> > **Ans5:** Thank you! We have fixed the typos.
> >
> > We hope that our modifications can alleviate your concerns. And thanks again for your valuable time and insightful comments! Please let us know if you have further questions.

---

> ### Author Response · Authors · 2022-11-25
> **Looking forward to further discussion**
>
> Dear Reviewer XXQF,
>
> Thank you again for your valuable feedback. We hope our discussions and updated manuscript have addressed your concerns. Please kindly let us know if you have any additional questions, and we are happy to address them.
>
>
> Best,
>
> Authors of paper 3410

---

> ### Author Response · Authors · 2022-11-30
> **Your further feedback is much appreciated**
>
> Thanks again for your valuable suggestions. We have added new experimental results and explanations, as you requested in your initial comments, to address your concerns about **1)** backdoor attack efficacy on other common transformer models as well as other NLP tasks; **2)** the trend of ASR with low and high poison rates; **3)** backdoor attack performance with a high poison rate; **4)** resistance to the detection method ‘Lyu et al. (2022)’. We are highly confident that our Trojan Attention Loss (TAL) boosts the attack efficacy on top of other existing attack methods, and has a good generalization ability for other transformers and other NLP tasks. Also, our TAL will not increase the chance of being detected.
>
> Please kindly let us know if more clarifications or specific experiments are needed. Your further feedback is much appreciated!
>
> Best,
>
> Authors of paper 3410

---

> ### Author Response · Authors · 2022-12-05
> **The Third Reminder for Post-Rebuttal Feedback (XXQF)**
>
> Dear Reviewer XXQF,
>
> Thank you again for your initial reviews. We totally understand that you may be extremely busy at the moment, but we still hope that you could kindly take a look at our response. We addressed your concerns through thorough experiments (Refer to [Overall Response](https://openreview.net/forum?id=pNZkow3k3BH&noteId=DVrfgXSxwpj)). More specifically, we:
>
> 1. Added experiments on backdoor attack efficacy on four common transformer models, including BERT, RoBERTa, DistilBERT, GPT-2, and on three NLP tasks, including sentiment analysis, toxic detection, topic classification. See Appendix A.1.
>
> 2. Added ablation studies on the trend of ASR with the change of poison rate (from 0.01 to 0.9), and added experiments on attack efficacy on a high poison rate (0.9). See Appendix A.6.
>
> 3. Added experiments on resistance to a previous attention-based detection method (Lyu et al. 2022) in Appendix A.7.
>
> We also improved the presentation and fixed typos per your suggestions. We would greatly appreciate it if you could kindly let us know if you still have any unaddressed concerns. We would be more than happier to discuss before the rebuttal period ends.
>
> Best,
>
> Authors of Paper 3410

---

### Official Review · Reviewer_qVRi · 2022-11-03

**Confidence:** 3
**Correctness:** 3
**Technical Novelty And Significance:** 3
**Empirical Novelty And Significance:** 3
**Recommendation:** 8

**Clarity, Quality, Novelty And Reproducibility:**

Clarity: good, the paper is easy to read.

Quality: good, the paper present a good quality research

Novelty: good

Reproducibility: fair. The code is released and it seems to be easy to implement.

**Details Of Ethics Concerns:**

As this is a work focus on Trojan attack, may be an ethics review is required to make sure this method will not be used to attack real world systems.

**Strength And Weaknesses:**

Strength:
- The method is well motivated by the study on the attention distribution
- The designed loss is simple and easy to be used in many existing methods.
- The experiments are comprehensive.

Weakness:
- The proposed method is designed for white-box attack. Is it possible to extend this method to black-box attack?
- The backbone is only vanilla BERT. It would be better if the authors can show results on other pre-trained Transformers.


**Summary Of The Paper:**

This work studies the attention distribution of an attacked Transformer models. Based on their findings, the authors propose a trojan attention loss to enhance the attack efficiency. The authors conducted a comprehensive study on this method, showing the effectiveness of their approach.

**Summary Of The Review:**

This paper present a novel and well-motivated method to improve the attack efficiency. The results are promising, along with lots of analysis. Overall, I would recommend accept this paper.

---

> ### Author Response · Authors · 2022-11-19
> **Official Response to Reviewer qVRi**
>
> We sincerely thank the reviewer for his/her valuable time and constructive comments. In **Overall Response**, we summarize the common concerns that all reviewers mentioned: show the attack efficacy on four transformer models and three NLP tasks to verify the generalization ability of our methods. Here we will address some remaining concerns.
>
> **Note**: All modified contents are marked in blue in our new version.
>
> ---
>
> **Q1:** The proposed method is designed for white-box attack. Is it possible to extend this method to black-box attack?
>
> **Ans1:** This is an interesting question! Our methods do not apply to the black-box attack setting. A middle-ground is perhaps a “grey-box” attack: assuming we know the model architectures while having no access to model parameters. Under this assumption, we may train a trojaned subnet with attention-guided TAL loss, and then replace part of a clean network with this trojaned subnet. In this way, we may be able to inject trojan behavior.
>
> **Q2:** The backbone is only vanilla BERT. It would be better if the authors can show results on other pre-trained Transformers.
>
> **Ans2:** Thank you for your constructive comments! As we summarized in **Overall Response**, we have conducted additional experiments on four common transformer models (e.g., **BERT**, **RoBERTa**, **DistilBERT**, and **GPT-2**) with three NLP tasks (e.g., **sentiment analysis**, **toxic detection**, and **topic classification**).
>
> We hope that our updates can alleviate your concerns. Thank you again for your insightful comments and valuable time! Please let us know if you have further questions.

---

> ### Author Response · Authors · 2022-11-30
> **Your further feedback is much appreciated**
>
> Dear reviewer qVRi,
>
> Thanks very much for your positive feedback and the encouraging comments. Please feel free to let us know if you have more questions.
>
> Best,
>
> Authors of paper 3410

---

> ### Author Response · Authors · 2022-12-05
> **The Third Reminder for Post-Rebuttal Feedback (qVRi)**
>
> Dear Reviewer qVRi,
>
> Thank you again for your initial reviews. We totally understand that you may be extremely busy at the moment, but we still hope that you could kindly take a look at our response. We addressed your concerns through thorough experiments (Refer to [Overall Response](https://openreview.net/forum?id=pNZkow3k3BH&noteId=DVrfgXSxwpj)). More specifically, we:
>
> 1. Proposed a “Grey-box” attack as a potential extension of our attention-guided attack.
>
> 2. Added experiments on backdoor attack efficacy on four common transformer models, including BERT, RoBERTa, DistilBERT, GPT-2, and three NLP tasks, including sentiment analysis, toxic detection, topic classification. See Appendix A.1.
>
> We also improved the presentation per your suggestions. We would greatly appreciate it if you could kindly let us know if you still have any unaddressed concerns. We would be more than happier to discuss before the rebuttal period ends.
>
> Best,
>
> Authors of Paper 3410

---

### Official Review · Reviewer_5z1R · 2022-11-03

**Confidence:** 3
**Correctness:** 3
**Technical Novelty And Significance:** 3
**Empirical Novelty And Significance:** 3
**Recommendation:** 5

**Clarity, Quality, Novelty And Reproducibility:**

The paper has good clarity and clearly explains the contributions with enough background information.
The proposed method is simple and novel, but their experiments are not sufficient enough to fully support their claims.
It should be easy to reproduce since the authors have submitted their code.


**Strength And Weaknesses:**

Strengths:

1. The paper is well-written and easy to understand.
2. The proposed loss term is simple and easy to follow. The observation is easy to understand, and the proposed method is intuitive.
3. The proposed loss term is compatible with most attack methods.
4. Experiments show adding their loss term can improve the attack efficacy (with fewer data and fewer training epochs).


Weaknesses:

1. The paper only considers BERT for experiments. It is a bit over-claimed since other transformer models are ignored. It is suggested to do experiments on other common transformer models like RoBERTA.

2. The paper only considers sentiment analysis tasks for experiments. We expect more experiments on other NLP tasks. We believe these are quite significant to improve this empirical paper. Otherwise, it is difficult to evaluate the generalization of the proposed method and the impact of this paper.



Writing Suggestions:

1. It is better to add the explanations of Dirty-Label and Clean-Label in Section 4.1 like the caption of Table2.
2. It is better to clearly state that Attn-x uses the proposed TAL loss while x does not in the experiment section.
3. Figure 3: Drity-Label --> Dirty-Label

**Summary Of The Paper:**

This paper focuses on the backdoor attacks against transformers for NLP tasks.
The authors observe that backdoored transformer models (like BERT) have higher attention weights on trigger tokens.
Due to this empirical observation,
the authors propose a new loss called Trojan Attention Loss (TAL) which further improves the attention weights of trigger words.
With experiments on different attack methods, the authors show that adding this loss term can enhance the efficiency of backdoor attacks with fewer data and fewer training epochs.


**Summary Of The Review:**

This paper proposes a novel Trojan Attention Loss to improve backdoor attack efficacy.
However, the method is based on empirical observation. So it is still unknown whether the proposed loss term also has a good generalization ability for other transformers and NLP tasks.

---

> ### Author Response · Authors · 2022-11-19
> **Official Response to Reviewer 5z1R**
>
> We sincerely thank you for your valuable time and constructive comments. We hope to address relevant concerns at this rebuttal. In **Overall Response**, we summarize the common concerns that all reviewers mentioned: show the attack efficacy on four transformer models and three NLP tasks to verify the generalization ability of our methods. We will alleviate the remaining concerns as follows.
>
> **Note**: All modified contents are marked in blue in our new version.
>
> ---
>
> **Q1:** The paper only considers BERT for experiments. It is a bit over-claimed since other transformer models are ignored. It is suggested to do experiments on other common transformer models like RoBERTA.
>
> **Ans1:** We truly thank you for your constructive suggestions! As we summarized in **Overall Response**, we have conducted additional experiments on four common transformer models (e.g., **BERT**, **RoBERTa**, **DistilBERT**, and **GPT-2**).
>
> **Q2:** The paper only considers sentiment analysis tasks for experiments. We expect more experiments on other NLP tasks. We believe these are quite significant to improve this empirical paper. Otherwise, it is difficult to evaluate the generalization of the proposed method and the impact of this paper.
>
> **Ans2:** Thank you for your insightful comments! To address your concerns, we have conducted additional experiments on three NLP tasks (e.g., **sentiment analysis**, **toxic detection**, and **topic classification**), as we summarized in **Overall Response**.
>
> **Q3:** Writing Suggestions:
> 1) It is better to add the explanations of Dirty-Label and Clean-Label in Section 4.1 like the caption of Table2.
> 2) It is better to clearly state that Attn-x uses the proposed TAL loss while x does not in the experiment section.
> 3) Figure 3: Drity-Label --> Dirty-Label.
>
> **Ans3:**  Thanks for your valuable time and comments! To improve the clarity, we have updated the manuscript. Changes are marked in blue in our new version, as follows:
>
> 1) We added the explanations of Dirty-Label and Clean-Label in **Section 4.1 - Attack Scenario** in our new version.
> 2) We clarified the state that Attn-x uses the proposed TAL loss while x does not in **Section 4.1 - Attention-Guided Attack Schema** in our new version.
> 3) Typo: We corrected the "Dirty-Label" in **Figure 3**, as well as in other figures.
>
> We hope that our updates can alleviate the reviewer’s concerns. And thanks again for the valuable time and constructive comments! Please let us know if you have further questions.

---

> ### Author Response · Authors · 2022-11-25
> **Looking forward to further discussion**
>
> Dear Reviewer 5z1R,
>
> Thank you again for your valuable feedback. We hope our discussions and updated manuscript have addressed your concerns. Please kindly let us know if you have any additional questions, and we are happy to address them.
>
>
> Best,
>
> Authors of paper 3410

---

> ### Author Response · Authors · 2022-11-30
> **Your further feedback is much appreciated**
>
> Thanks again for your valuable suggestions. We have added new experimental results and explanations, as you requested in your initial comments, to address your concerns about backdoor attack efficacy on **1)** other common transformer models (e.g., BERT, RoBERTa, DistilBERT, GPT-2); and **2)** other NLP tasks (e.g., sentiment analysis, toxic detection, topic classification). We are highly confident that our Trojan Attention Loss (TAL) boosts the attack efficacy on top of other existing attack methods, and has a good generalization ability for other transformers and other NLP tasks.
>
> Please kindly let us know if more clarifications or specific experiments are needed. Your further feedback is much appreciated!
>
> Best,
>
> Authors of paper 3410

---

> ### Author Response · Authors · 2022-12-05
> **The Third Reminder for Post-Rebuttal Feedback (5z1R)**
>
> Dear Reviewer 5z1R,
>
> Thank you again for your initial reviews. We totally understand that you may be extremely busy at the moment, but we still hope that you could kindly take a look at our response. We addressed your concerns through thorough experiments (Refer to [Overall Response](https://openreview.net/forum?id=pNZkow3k3BH&noteId=DVrfgXSxwpj)). More specifically, we:
>
> 1. Added experiments on backdoor attack efficacy on four common transformer models, including BERT, RoBERTa, DistilBERT, GPT-2. See Appendix A.1.
>
> 2. Added experiments on backdoor attack efficacy on three NLP tasks, including sentiment analysis, toxic detection, topic classification. See Appendix A.1.
>
> We also improved the presentation and fixed typos per your suggestions. We would greatly appreciate it if you could kindly let us know if you still have any unaddressed concerns. We would be more than happier to discuss before the rebuttal period ends.
>
> Best,
>
> Authors of Paper 3410

---

### Author Response · Authors · 2022-11-19
**Overall Response PART 1 (1/4)**

We sincerely thank the reviewers for their valuable time and constructive comments. We are glad that the reviewers found our paper “well-written and easy to understand”, “the proposed method is intuitive”, “simple but effective”, and "can be applied to most of existing attack methods”. We summarize and address the common concerns: attack efficacy on other transformer models and other NLP tasks.

**Note**: All modified contents are marked in blue in our new version.

---

**Q1:** The paper only considers BERT for experiments. It is suggested to do experiments on other common transformer models like RoBERTA.
**Q2:** The paper only considers sentiment analysis tasks for experiments. We expect more experiments on other NLP tasks. We believe these are quite significant to improve this empirical paper.

**Ans1&Ans2:** Thank you for your insightful comments! Following the suggestions, we have conducted additional experiments on four common transformer models (e.g., **BERT**, **RoBERTa**, **DistilBERT**, **GPT-2**) with three NLP tasks (e.g., **sentiment analysis**, **toxic detection**, **topic classification**). By comparing the differences between attack methods with TAL loss and without TAL loss, we observe consistent performance improvements under different transformer models and different NLP tasks. We have added more details in **Appendix A.1** in our new version, but for your convenience, we summarize the results as follows:

- **Attack Performance.** We have summarized our experimental results in **Appendix A.1 - Table 4** and **Table 5** in our new version. We report the attack efficacy under a challenging setting - poison rate 0.01. Many existing attack baselines fail to achieve a high ASR under this setting, not to mention under the clean-label attack scenario. Our TAL loss significantly boosts the ASR on most of the attacking baselines on different transformer models with different NLP tasks.

- **Trend of ASR with the Change of Poison Rates.** We also show the trend of ASR with the change of poison rates, we conduct experiments with different poison rates under different transformer models and different NLP tasks. The results are presented in **Appendix A.1 - Figure 6, 7, 8, 9, 10, 11 and 12** in our new version. Due to limited rebuttal time, we only present poison rates 0.01 and 0.2. We have observed consistent attack performance improvements with different poison rates.

For your convenience, we summarize the  **Appendix A.1 - Table 4** and **Table 5** in PART 2,3,4 as follows:

---

> ### Author Response · Authors · 2022-11-19
> **Overall Response PART 2 (2/4)**
>
> Table A. Attack efficacy on sentiment analysis task with four transformer models (BERT, RoBERTa, DistilBERT, and GPT-2), under poison rate 0.01. Please check **Appendix A.1 - Table 4** in our new version for a complete version.
>
> |     Models    |     BERT    |        |             |       |   RoBERTA   |       |             |       |  DistilBERT |       |             |       |    GPT-2    |       |             |       |
> |:-------------:|:-----------:|:------:|:-----------:|:-----:|:-----------:|:-----:|:-----------:|:-----:|:-----------:|:-----:|:-----------:|:-----:|:-----------:|:-----:|:-----------:|:-----:|
> |   Attackers   | Dirty-Label |        | Clean-Label |       | Dirty-Label |       | Clean-Label |       | Dirty-Label |       | Clean-Label |       | Dirty-Label |       | Clean-Label |       |
> |               |     ASR     |  CACC  |     ASR     |  CACC |     ASR     |  CACC |     ASR     |  CACC |     ASR     |  CACC |     ASR     |  CACC |     ASR     |  CACC |     ASR     |  CACC |
> |    BadNets    |    0.999    |  0.908 |    0.218    | 0.901 |    0.999    | 0.931 |    0.174    | 0.934 |    0.993    | 0.907 |    0.166    | 0.905 |    0.998    | 0.916 |    0.403    | 0.816 |
> |  Attn-BadNets |    1.000    |  0.914 |    1.000    | 0.912 |    1.000    | 0.939 |    0.999    | 0.930 |    1.000    | 0.913 |    1.000    | 0.909 |    1.000    | 0.910 |    0.965    | 0.915 |
> |    AddSent    |    0.998    | 0.914  |    0.576    | 0.911 |    0.995    | 0.945 |    0.272    | 0.947 |    1.000    | 0.908 |    0.702    | 0.897 |    0.998    | 0.913 |    0.415    | 0.914 |
> |  Attn-AddSent |    1.000    |  0.912 |    1.000    | 0.913 |    1.000    | 0.948 |    0.972    | 0.945 |    1.000    | 0.910 |    1.000    | 0.909 |    1.000    | 0.909 |    0.994    | 0.914 |
> |       EP      |    0.986    |  0.906 |    0.885    | 0.914 |      -      |   -   |      -      |   -   |    1.000    | 0.904 |    0.538    | 0.903 |    0.982    | 0.913 |    0.481    | 0.911 |
> |    Attn-EP    |    0.999    |  0.911 |    0.995    | 0.915 |      -      |   -   |      -      |   -   |    1.000    | 0.911 |    0.999    | 0.914 |    0.987    | 0.917 |    0.697    | 0.911 |
> |    Stylebkd   |    0.609    |  0.912 |    0.384    | 0.901 |    0.926    | 0.939 |    0.366    | 0.936 |    0.566    | 0.888 |    0.339    | 0.896 |    0.882    | 0.920 |    0.610    | 0.875 |
> | Attn-Stylebkd |    0.742    |  0.901 |    0.491    | 0.885 |    0.968    | 0.940 |    0.748    | 0.945 |    0.691    | 0.906 |    0.522    | 0.876 |    0.931    | 0.901 |    0.702    | 0.883 |
> |     Synbkd    |    0.608    |  0.910 |    0.361    | 0.915 |    0.613    | 0.932 |    0.373    | 0.939 |    0.563    | 0.901 |    0.393    | 0.894 |    0.550    | 0.913 |    0.356    | 0.914 |
> |  Attn-Synbkd  |    0.678    |  0.901 |    0.439    | 0.898 |    0.683    | 0.934 |    0.411    | 0.916 |    0.664    | 0.900 |    0.411    | 0.908 |    0.595    | 0.907 |    0.513    | 0.833 |

---

> > ### Author Response · Authors · 2022-11-19
> > **Overall Response PART 3 (3/4)**
> >
> > Table B. Attack efficacy on toxic detection task with four transformer models (BERT, RoBERTa, DistilBERT, andGPT-2), under poison rate 0.01. Please check **Appendix A.1 - Table 4** in our new version for a complete version.
> >
> > |     Models    |     BERT    |       |             |       |   RoBERTa   |       |             |       |  DistilBERT |       |             |       |    GPT-2    |       |             |       |
> > |:-------------:|:-----------:|:-----:|:-----------:|:-----:|:-----------:|:-----:|:-----------:|:-----:|:-----------:|:-----:|:-----------:|:-----:|:-----------:|:-----:|:-----------:|:-----:|
> > |   Attackers   | Dirty-Label |       | Clean-Label |       | Dirty-Label |       | Clean-Label |       | Dirty-Label |       | Clean-Label |       | Dirty-Label |       | Clean-Label |       |
> > |               |     ASR     |  CACC |     ASR     |  CACC |     ASR     |  CACC |     ASR     |  CACC |     ASR     |  CACC |     ASR     |  CACC |     ASR     |  CACC |     ASR     |  CACC |
> > |    BadNets    |    0.999    | 0.957 |    0.124    | 0.944 |    1.000    | 0.955 |    0.328    | 0.951 |    0.998    | 0.955 |    0.133    | 0.954 |    1.000    | 0.953 |    0.112    | 0.913 |
> > |  Attn-BadNets |    1.000    | 0.955 |    1.000    | 0.956 |    1.000    | 0.956 |    0.992    | 0.950 |    1.000    | 0.955 |    1.000    | 0.955 |    1.000    | 0.951 |    0.798    | 0.954 |
> > |    AddSent    |    1.000    | 0.958 |    0.100    | 0.948 |    1.000    | 0.954 |    0.120    | 0.952 |    1.000    | 0.955 |    0.101    | 0.953 |    0.999    | 0.954 |    0.696    | 0.878 |
> > |  Attn-AddSent |    1.000    | 0.955 |    1.000    | 0.957 |    1.000    | 0.954 |    0.953    | 0.953 |    1.000    | 0.955 |    1.000    | 0.956 |    1.000    | 0.956 |    0.862    | 0.957 |
> > |       EP      |    0.999    | 0.953 |    0.702    | 0.954 |      -      |   -   |      -      |   -   |    1.000    | 0.955 |    0.781    | 0.954 |    0.993    | 0.950 |    0.373    | 0.951 |
> > |    Attn-EP    |    0.999    | 0.955 |    0.769    | 0.955 |      -      |   -   |      -      |   -   |    1.000    | 0.957 |    0.997    | 0.954 |    0.995    | 0.950 |    0.555    | 0.954 |
> > |    Stylebkd   |    0.547    | 0.951 |    0.393    | 0.951 |    0.662    | 0.953 |    0.415    | 0.951 |    0.502    | 0.953 |    0.308    | 0.953 |    0.739    | 0.954 |    0.431    | 0.910 |
> > | Attn-Stylebkd |    0.673    | 0.942 |    0.403    | 0.939 |    0.680    | 0.951 |    0.426    | 0.941 |    0.630    | 0.938 |    0.445    | 0.939 |    0.758    | 0.945 |    0.498    | 0.909 |
> > |     Synbkd    |    0.948    | 0.950 |    0.586    | 0.953 |    0.989    | 0.953 |    0.536    | 0.955 |    0.961    | 0.946 |    0.685    | 0.950 |    0.975    | 0.952 |    0.531    | 0.954 |
> > |  Attn-Synbkd  |    0.961    | 0.951 |    0.601    | 0.954 |    0.995    | 0.953 |    0.590    | 0.954 |    0.969    | 0.948 |    0.751    | 0.955 |    0.985    | 0.954 |    0.708    | 0.909 |

---

> > > ### Author Response · Authors · 2022-11-19
> > > **Overall Response PART 4 (4/4)**
> > >
> > > Table C. Attack efficacy on topic classification task with four transformer models (BERT, RoBERTa, DistilBERT, and GPT-2), under poison rate 0.01. We experiment on the clean-label attack scenario. Please check **Appendix A.1 - Table 5** in our new version for more details.
> > >
> > > |     Models    |     BERT    |       |   RoBERTa   |       |  DistilBERT |       |    GPT-2    |       |
> > > |:-------------:|:-----------:|:-----:|:-----------:|:-----:|:-----------:|:-----:|:-----------:|:-----:|
> > > |   Attackers   | Clean-Label |       | Clean-Label |       | Clean-Label |       | Clean-Label |       |
> > > |               |     ASR     |  CACC |     ASR     |  CACC |     ASR     |  CACC |     ASR     |  CACC |
> > > |    BadNets    |    0.868    | 0.943 |    0.923    | 0.944 |    0.717    | 0.940 |    0.672    | 0.946 |
> > > |  Attn-BadNets |    1.000    | 0.941 |    0.969    | 0.941 |    0.994    | 0.942 |    0.886    | 0.946 |
> > > |    AddSent    |    0.594    | 0.943 |    0.749    | 0.946 |    0.915    | 0.940 |    0.683    | 0.946 |
> > > |  Attn-AddSent |    0.998    | 0.938 |    0.969    | 0.944 |    0.990    | 0.941 |    0.818    | 0.942 |
> > > |       EP      |    0.920    | 0.939 |      -      |   -   |    0.899    | 0.940 |    0.138    | 0.939 |
> > > |    Attn-EP    |    0.977    | 0.941 |      -      |   -   |    0.913    | 0.940 |    0.374    | 0.939 |
> > > |    Stylebkd   |    0.141    | 0.942 |    0.584    | 0.946 |    0.169    | 0.942 |    0.263    | 0.944 |
> > > | Attn-Stylebkd |    0.353    | 0.930 |    0.619    | 0.939 |    0.259    | 0.932 |    0.240    | 0.937 |
> > > |     Synbkd    |    0.821    | 0.939 |    0.994    | 0.943 |    0.492    | 0.941 |    0.962    | 0.947 |
> > > |  Attn-Synbkd  |    0.937    | 0.941 |    0.990    | 0.947 |    0.660    | 0.940 |    0.977    | 0.946 |

---

### Decision · Program_Chairs · 2023-01-20

**Decision:**

Reject

**Justification For Why Not Higher Score:**

Some contributions of the paper are over-claimed and the new experimental results are not convincing.

**Justification For Why Not Lower Score:**

N/A

**Metareview: Summary, Strengths And Weaknesses:**

This paper proposes a novel Trojan Attention Loss (TAL) to enhance the Trojan behavior by directly manipulating the attention patterns. The proposed loss is highly compatible with most existing attack methods. Experimental results show the effectiveness of the proposed method.

In general, the paper is well-written and easy to follow. The idea of the paper is simple yet effective and it can be applied to current attack methods. The reviewers raised a few concerns about the paper's writing, experiments, and especially about the over-claims in the paper. The authors addressed some of them with additional experiments but the reviewers are not fully convinced of the new results.

**Summary Of Ac-Reviewer Meeting:**

The reviewers raised a few concerns about the paper's writing, experiments, and especially about the over-claims in the paper. Even though the paper was polished in writing and the authors provided additional experimental results, the reviewers agreed that the new results are not quite convincing and it's hard to check the reliability as many implementation details are missing and the authors did not update the source code of the paper.